# Symbal: Detecting Systematic Misalignments in Model-Generated Captions

**Maya Varma** [1]   **Jean-Benoit Delbrouck** [1 2]   **Sophie Ostmeier** [1]   **Akshay Chaudhari** [* 1]   **Curtis Langlotz** [* 1]

## Abstract

Multimodal large language models (MLLMs) often introduce errors when generating image captions, resulting in misaligned image-text pairs. Our work focuses on a class of captioning errors that we refer to as *systematic misalignments*, where a recurring error in MLLM-generated captions is closely associated with the presence of a specific visual feature in the paired image. Given a vision-language dataset with MLLM-generated captions, our aim in this work is to detect such errors, a task we refer to as systematic misalignment detection. As our first key contribution, we present SYMBAL, which utilizes a structured, dual-stage setup with off-the-shelf foundation models to identify systematic misalignments and summarize results in natural language. As our second key contribution, we introduce SYMBAL-BENCH, a benchmark designed to evaluate automated methods on our proposed task. SYMBAL-BENCH consists of 1.7 million image-text pairs from two domains (natural and medical images), organized into 420 vision-language datasets with annotated systematic misalignments. SYMBAL exhibits strong performance on this benchmark, correctly identifying systematic misalignments in 63.8% of datasets, a nearly 4x improvement over the closest baseline. We supplement our evaluations on SYMBALBENCH with real-world evaluations, showing that (1) SYMBAL can accurately surface systematic misalignments in captions generated by four MLLMs and (2) SYMBAL is a powerful tool for auditing off-the-shelf image-caption datasets. Ultimately, our novel task, method, and benchmark can aid users with auditing MLLM-generated captions and identifying critical errors, without requiring access to the underlying MLLM. Code is available at https://github.com/Stanford-AIMI/Symbal.

*Equal senior authorship  [1]Stanford University [2]HOPPR. Correspondence to: Maya Varma <mayavarma@cs.stanford.edu>.

*Proceedings of the 43rd International Conference on Machine Learning*, Seoul, South Korea. PMLR 306, 2026. Copyright 2026 by the author(s).

## 1. Introduction

Multimodal large language models (MLLMs) possess strong image captioning capabilities yet often introduce errors into generated captions (Sarto et al., 2025; Zhou et al., 2024; Liu et al., 2025). As a result, images and paired MLLM-generated captions may be *misaligned*, meaning that the generated text erroneously refers to features that are not visible in the image. For example, consider an MLLM that is tasked with generating a radiology report for an input medical image; in this setting, a misalignment may exist if the MLLM-generated report indicates the presence of cardiomegaly (a condition characterized by an enlarged heart) despite the image showing no evidence of this diagnosis. Misalignments can have severe consequences, particularly in safety-critical domains like medicine (Hardy et al., 2025; Nakaura et al., 2023).

Our work focuses on a critical yet previously-underexplored subclass of captioning errors that we refer to as *systematic misalignments*. We term a misalignment as *systematic* when a recurring error in MLLM-generated captions is closely associated with the presence of a specific visual feature in the paired image. For example, in the medical domain, incorrect diagnoses of cardiomegaly in the MLLM-generated reports may be strongly associated with the presence of pacemakers (an implanted medical device that regulates the heartbeat) in the corresponding image (Sourget et al., 2025; Kumar et al., 2025). Systematic misalignments are a particularly egregious class of errors because they often arise due to spurious correlations or biases learned by MLLMs during training. As a result, systematic misalignments typically involve features that frequently co-occur in the real-world yet are not deterministically linked; for instance, while cardiomegaly and pacemakers do co-occur frequently, the presence of a pacemaker in a medical image does not necessarily imply that the patient has cardiomegaly. Thus, errors associated with systematic misalignments may seem highly plausible and are consequently challenging to detect.

In this work, we introduce the *systematic misalignment detection* task with the goal of leveraging automated approaches to identify this challenging class of captioning errors. A method that aims to solve the systematic misalignment detection task will accept as input a vision-language dataset, which consists of images paired with free-form

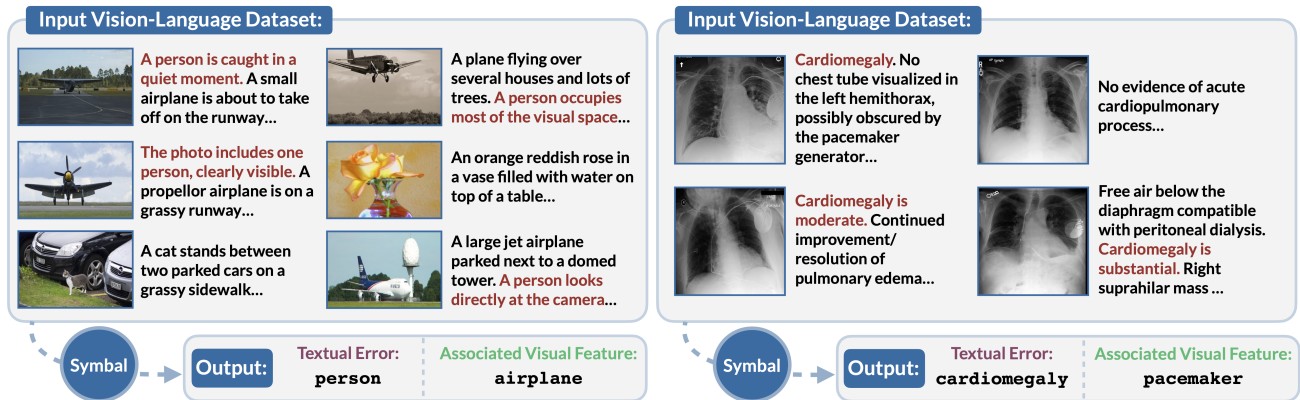

*Figure 1.* Given an input dataset with thousands of images and paired MLLM-generated captions, the *systematic misalignment detection task* involves identifying recurring textual errors and associated visual features. Here, we provide example image-caption pairs from two datasets in SYMBALBENCH with expected outputs.

MLLM-generated captions. Then, as output, the method must identify textual errors (e.g. "cardiomegaly" in the previous example) that are systematically associated with visual features (e.g. "pacemaker" in the previous example).

Addressing the systematic misalignment detection task with automated methods is challenging for the following two reasons. First, vision-language datasets provided as input to automated methods are often large in size with thousands of image-caption pairs; identifying global error patterns from such datasets is nontrivial, especially since the size of such datasets exceeds the reasoning capabilities of even state-of-the-art models. Second, there are no existing benchmarks for comprehensively evaluating methods on their ability to discover systematic misalignments. In order to address these challenges, we present the following contributions:

- We propose SYMBAL, an automated approach for detecting systematic misalignments in MLLM-generated captions.[1] Our key insight is to structure the systematic misalignment detection task into two stages, with each stage comprised of individual subtasks. The first stage of SYMBAL focuses solely on identifying recurring textual errors in captions; to this end, SYMBAL clusters textual facts based on semantic similarity, scores each cluster by degree of misalignment with paired images, and summarizes the top-ranked cluster into a single unifying concept. The second stage of SYMBAL then leverages this information to identify and describe the associated visual feature.

- We introduce SYMBALBENCH, the first benchmark designed to evaluate automated methods for systematic misalignment detection. SYMBALBENCH consists of 420 image-caption datasets, each paired with a ground-truth

label for a systematic misalignment. Methods are then quantitatively evaluated on the extent to which their predictions align with the ground truth.

We evaluate SYMBAL using SYMBALBENCH, analyzing a range of approaches for each subtask. The best configuration of SYMBAL correctly identifies the systematic misalignment in 63.8% of SYMBALBENCH datasets. SYMBAL exhibits a nearly 4x improvement over the closest baseline, demonstrating the utility of our dual-stage, structured approach for addressing the systematic misalignment detection task. Finally, we supplement our evaluations on SYMBALBENCH with real-world evaluations, demonstrating quantitatively and qualitatively that (1) SYMBAL can accurately surface systematic misalignments in captions generated by four MLLMs and (2) SYMBAL is a powerful tool for auditing off-the-shelf datasets with MLLM-generated captions.

Ultimately, we envision our novel task, benchmark, and method aiding in the following real-world contexts. First, our approach reveals insights into failure modes of trained MLLMs, which can (1) provide developers with critical information for building more robust models as well as (2) assist end-users with understanding limitations prior to real-world deployment. For instance, returning to our previous example, physicians using an MLLM in the clinic can be forewarned that generated reports tend to incorrectly diagnose "cardiomegaly" when X-rays have visible "pacemakers"; knowledge of this failure mode can allow for further manual review of model outputs on those cases. Second, our approach can help users identify systematic captioning errors in off-the-shelf datasets, even *in black-box settings where access to the underlying MLLM is unavailable*. This is a particularly important use-case, especially as publicly-available image datasets with MLLM-generated captions become widely used for training the next generation of multimodal foundation models.

---

[1]The acronym SYMBAL refers to **sy**stematic **m**isalignment detection **b**etween im**a**ges and **l**anguage.

**Conflict of Interest Disclosure.** None. Funding sources are listed in the Acknowledgments at the end of this paper.

## 2. Related Work

Our work builds on three prior lines of study: (1) *local misalignment detection* methods that identify captioning errors at the per-sample level; (2) *global error detection* methods that summarize systematic trends in prediction errors; and (3) methods for *describing patterns in large datasets with natural language*.

**Local Misalignment Detection:** Given a single image and its paired model-generated caption, one line of recent work has focused on developing metrics that measure image-caption alignment using numeric scores. Examples include reference-free metrics like CLIPScore (Hessel et al., 2021) and PAC-S (Sarto et al., 2023), which do not require the existence of ground-truth captions; on the other hand, reference-based metrics such as BLEU (Papineni et al., 2002), ROUGE (Lin, 2004), CIDEr (Vedantam et al., 2015), METEOR (Banerjee & Lavie, 2005), and Ref-CLIPScore (Hessel et al., 2021) make use of ground-truth captions. The utility of such metrics is typically evaluated using image-caption benchmarks with human-annotated quality judgments (e.g. FLICKR8K-Expert (Hodosh et al., 2013), Pascal-50S (Vedantam et al., 2015), ReXVal (Yu et al., 2023)) or known model-injected errors (e.g. FOIL (Shekhar et al., 2017), ReXErr (Rao et al., 2025)).

Several recent works have extended numeric scoring strategies by proposing interpretable metrics, which are capable of identifying the specific features in model-generated captions that are incorrect with respect to the image. Examples include reference-based metrics like CHAIR (Rohrbach et al., 2018), ALOHa (Petryk et al., 2024), and GREEN (Ostmeier et al., 2024) as well as reference-free metrics like FLEUR (Lee et al., 2024). Our work draws inspiration from these studies by also prioritizing interpretability; our method SYMBAL not only detects whether captioning errors are present but also provides users with a natural language output indicating the erroneous textual facts and associated visual cues. However, our study exhibits a key distinction from this line of work: whereas these metrics evaluate a single image and its paired model-generated caption, our work instead focuses on detecting *global*, systematic trends in captioning errors.

**Global Error Detection:** Due to visual biases or spurious correlations learned during training, machine learning models often make systematic prediction errors at test time. Selected examples in the classification setting noted by prior works include (1) an object recognition model that can correctly classify cows in pastoral settings yet demonstrates high error rates when cows are in beach settings (Beery et al., 2018) and (2) a pneumothorax detection model that achieves radiologist-level overall accuracy yet demonstrates high error rates when chest tubes, a medical device used for treatment, are absent (Oakden-Rayner et al., 2020). Detecting such failures is challenging due to the fact that relevant subgroups are typically not annotated in data.

A recent line of work has explored the development of automated methods for identifying global, systematic error patterns in classification settings. Given a validation dataset with images, model predictions, and ground-truth labels, these methods identify specific visual features (e.g. the beach background or the absence of tubes in the above examples) that are associated with higher error rates (Eyuboglu et al., 2022; Jain et al., 2023; Sohoni et al., 2020; Varma et al., 2024). Our work shares a similar goal in identifying systematic error patterns; however, we extend beyond the classification setting to the image captioning setting, where input datasets consist of images and paired model-generated captions. The inclusion of free-form text in input datasets presents an added level of complexity in comparison to labels. Additionally, we explicitly consider settings where ground-truth captions are unavailable.

**Describing Datasets with Natural Language:** Several works have presented approaches for describing patterns in large datasets using natural language (Burgess et al., 2025). In particular, recent studies have generated natural language descriptions (i) summarizing differences given two input datasets (Dunlap et al., 2024; Zhong et al., 2022) and (ii) summarizing model prediction errors given classification datasets with labels (Eyuboglu et al., 2022; Menon & Srivastava, 2024; Kim et al., 2024). Our work also involves summarizing dataset-level patterns with natural language; however, in our setting, datasets consist of images and paired captions, and descriptions must specifically identify systematic misalignments.

## 3. Task Definition

In this section, we formally introduce the systematic misalignment detection task. Consider a vision-language dataset $\mathcal{D} = \{(V_i, T_i)\}_{i=1}^{N}$ consisting of images $V$ paired with free-form, model-generated text $T$. For example, dataset $\mathcal{D}$ may consist of chest X-rays $V$ paired with MLLM-generated radiology reports $T$. We will express each text sample $T_i$ as a collection of textual facts $T_i = \{t_1^i, t_2^i, ..., t_{n_i}^i\}$ and each image $V_i$ as a collection of visual features $V_i = \{v_1^i, v_2^i, ..., v_{m_i}^i\}$.

Dataset $\mathcal{D}$ may include misaligned samples, where text $T_i$ does not accurately describe the content of the paired image $V_i$. We consider a pair $(V_i, T_i)$ to be misaligned if there exists at least one erroneous textual fact $t_k^i \in T_i$ that does not accurately describe any visual feature $v_j^i \in V_i$. Mis-

alignments are particularly egregious when they occur in a *systematic* fashion, meaning that an erroneous textual fact $t$ is repeatedly associated with the presence of a visual feature $v$ throughout a dataset. For instance, in the medical imaging example discussed earlier, incorrect diagnoses of cardiomegaly in MLLM-generated reports are strongly associated with the presence of a pacemaker in the corresponding chest X-rays; this suggests the existence of a systematic misalignment between reports containing $t = cardiomegaly$ and images containing $v = pacemaker$.

Thus, given $\mathcal{D}$, the goal of the **systematic misalignment detection** task is to discover textual errors $t$ that are systematically associated with visual cues $v$. A method $\mathcal{M} : \mathcal{D} \rightarrow (\hat{t}, \hat{v})$ that aims to solve the systematic misalignment detection task will accept dataset $\mathcal{D}$ as input; we note here that datasets may be large in size, consisting of thousands of image-text pairs. Then, method $\mathcal{M}$ will predict $(\hat{t}, \hat{v})$ as output, indicating the discovered textual error $\hat{t}$ and associated visual feature $\hat{v}$; here, both $\hat{t}$ and $\hat{v}$ will be expressed in text.

We consider two variants of input dataset $\mathcal{D}$: (1) *reference-free*, where each sample in dataset $\mathcal{D} = \{(V_i, T_i)\}_{i=1}^N$ consists of image $V_i$ and model-generated text $T_i$, and (2) *reference-based*, where each sample in dataset $\mathcal{D} = \{(V_i, T_i, R_i)\}_{i=1}^N$ consists of an image $V_i$, model-generated text $T_i$, and a ground-truth reference caption $R_i$.

## 4. Our Approach: SYMBAL

The systematic misalignment detection task is made challenging by the fact that vision-language datasets may be complex and large in size; identifying global error patterns from such datasets is nontrivial. In this section, we address this challenge with our approach SYMBAL, which structures the systematic misalignment detection task into two stages. Each stage is comprised of three individual subtasks: grouping, scoring, and summarizing. Sections 4.1 and 4.2 discuss the two stages in detail.

### 4.1. Stage 1: Detecting Erroneous Textual Facts

The first stage of SYMBAL predicts the erroneous textual fact by (1) grouping semantically-similar facts that occur consistently throughout the dataset, (2) scoring each group of facts by degree of misalignment with paired images, (3) and summarizing the top-ranked group of facts into a single unifying concept $\hat{t}$. The three subtasks associated with Stage 1 are detailed below:

- **Grouping semantically-similar facts:** As defined in Section 3, we first express each text sample $T_i$ as a collection of textual facts $T_i = \{t_1^i, t_2^i, ..., t_{n_i}^i\}$ by splitting captions at the sentence level. We then identify clusters of semantically-similar facts that occur in $\mathcal{D}$; for example, in

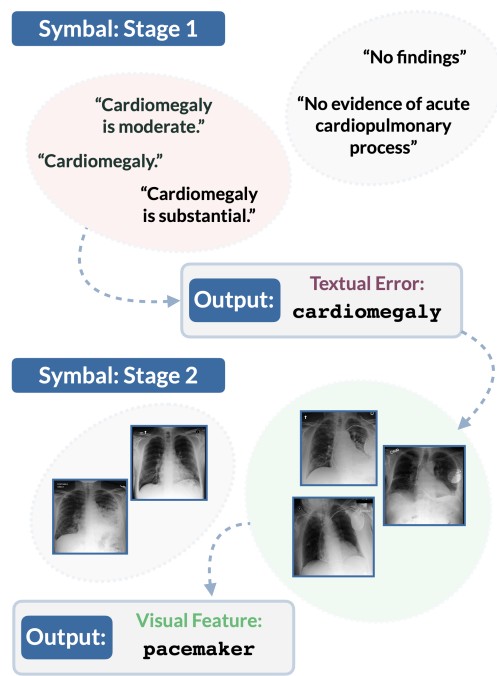

*Figure 2.* SYMBAL detects systematic misalignments with a two-stage procedure. The first stage involves detecting erroneous textual facts, and the second stage involves detecting associated visual features.

the medical imaging example discussed earlier, perhaps one such cluster will contain sentences from radiology reports that discuss the presence of cardiomegaly. To this end, we aggregate all textual facts in $\mathcal{D}$, forming the set $\bigcup_{i=1}^N T_i = \{t_k^i : i = 1, ..., N; k = 1, ..., n_i\}$. Each textual fact in this set is encoded using a text embedding model; then, embeddings are clustered using spherical K-Means, where the number of clusters is selected automatically using Silhouette distance.

- **Scoring groups by degree of misalignment:** Next, we score each cluster by computing the mean degree of alignment between constituent textual facts and paired images. Based on methods from prior work (Hessel et al., 2021; Dunlap et al., 2024; Chen et al., 2024a), we consider three options for measuring alignment between a given textual fact and its paired image: (1) *embedding scorer*, which computes embeddings for the text and image modalities and measures alignment as the cosine similarity, (2) *text-only scorer*, which generates a caption for the image and tasks an LLM with determining if the textual fact is accurate with respect to the caption, and (3) *vision-language scorer*, where a MLLM is provided both the image and the textual fact as input and tasked with determining if the textual fact is accurate. Low scores suggest that a large proportion of textual facts in the cluster are misaligned with respect to their paired images.

- **Summarizing the top-ranked group:** Given the alignment scores computed in the previous step, we identify the cluster exhibiting the highest degree of misalignment, which we refer to as $C_{text}$. Then, we apply a *text-only summarizer*, where an LLM is provided a list of textual facts in $C_{text}$ and tasked with identifying the unifying concept.

The final output of the summarizer is the predicted erroneous textual fact $\hat{t}$; for example, in the medical example discussed earlier, the predicted textual fact may be $\hat{t} = cardiomegaly$. In Section 6.1, we evaluate the role of various text embedding models and alignment scorers.

### 4.2. Stage 2: Detecting Associated Visual Features

We now proceed to the second stage of SYMBAL, which predicts the associated visual feature by (1) grouping semantically-similar images paired with text containing fact $\hat{t}$, (2) scoring each group of images by degree of misalignment with $\hat{t}$, and (3) summarizing the top-ranked group of images into a single unifying concept $\hat{v}$. The three subtasks associated with Stage 2 are detailed below:

- **Grouping semantically-similar images:** We begin by identifying all images $V_i \in \mathcal{D}$ containing at least one paired textual fact in cluster $C_{text}$ (i.e. where $t_k^i \in C_{text}$ for some $k$). Each image in this set is encoded using an image embedding model; then, embeddings are clustered using spherical K-Means, where the number of clusters is selected automatically using Silhouette distance.

- **Scoring groups by degree of misalignment:** Next, we score each cluster by computing the mean degree of misalignment between images and paired textual facts in $C_{text}$. We consider the same scoring mechanisms as in Stage 1. Low scores suggest that a large proportion of images in the cluster are misaligned with fact $\hat{t}$.

- **Summarizing the top-ranked group:** Given the alignment scores computed in the previous step, we identify the cluster exhibiting the highest degree of misalignment, which we will refer to as $C_{image}$. Then, we consider two summarization mechanisms for identifying the unifying concept shared by images in $C_{image}$: (1) *text-only summarizer*, where a caption is generated for each image in $C_{image}$ and an LLM is tasked with identifying the unifying concept, and (2) *vision-language summarizer*, where an MLLM is provided with images in $C_{image}$ and tasked with identifying the unifying concept.

The final output of the summarizer is the predicted visual feature $\hat{v}$; for example, in the medical example discussed earlier, the predicted visual feature may be $\hat{v} = pacemaker$.

In Section 6.2, we evaluate the role of various image embedding models, alignment scorers, and summarizers.

We note here that some datasets may contain multiple systematic misalignments; SYMBAL can be trivially extended to such settings, as we show in Appendix A and E.

## 5. Benchmark: SYMBALBENCH

The key challenge behind evaluating methods like SYMBAL on real-world vision-language datasets is that ground-truth systematic misalignments are typically unknown. Moreover, collecting human annotations for a task at this scale, where datasets include thousands of images paired with information-dense captions, is simply intractable. Thus, without access to ground-truth annotations, it becomes difficult (1) to determine whether misalignments identified by a method like SYMBAL are accurate and (2) to quantitatively compare results across multiple methods.

In this section, we introduce SYMBALBENCH, which is designed to address this challenge. Specifically, SYMBALBENCH utilizes an automated method to inject a pre-defined systematic misalignment into a base vision-language dataset, yielding an evaluation setting where a ground-truth annotation $(t, v)$ is available. The automated nature of our approach provides several key advantages, including (1) the ability to generate hundreds of evaluation settings simply by injecting varied systematic misalignments, (2) the presence of ground-truth labels that are guaranteed to be accurate, and (3) the ability to extend to specialized domains like medical imaging. In Section 6.4, we augment our evaluations on SYMBALBENCH with real-world analyses.

**Benchmark Design:** SYMBALBENCH consists of 420 *evaluation settings*, where each setting is comprised of a vision-language dataset $\mathcal{D}$ and an associated ground-truth label $(t,v)$ representing the systematic misalignment. In order to create each evaluation setting, we (1) obtain a high-quality base dataset with images and paired text, (2) predefine a systematic misalignment $(t, v)$, and (3) inject the erroneous textual fact $t$ into the base dataset such that a strong association exists with visual feature $v$. Below, we discuss these three steps in detail:

1. **Obtaining a base dataset.** We begin by obtaining an off-the-shelf vision-language dataset with high-quality samples. We consider two options for the base dataset: COCO (2017 val split) (Lin et al., 2014) and MIMIC-CXR (test split) (Johnson et al., 2019a). COCO consists of natural images depicting common objects from 80 categories. After preprocessing, the base dataset includes a total of 4349 images with associated captions. MIMIC-CXR consists of chest X-rays and associated radiology reports obtained from the Beth Israel Deaconess Medical Center. After preprocessing, the base dataset includes

2233 images, each paired with the "Impressions" section of the corresponding report.

2. **Predefining a systematic misalignment.** Given a base dataset, we predefine a systematic misalignment consisting of a textual fact $t$ and associated visual feature $v$. Predefined misalignments are meant to emulate those that are likely to emerge when using real-world, off-the-shelf MLLMs to generate captions. For COCO, we sample $t$ and $v$ from the set of 80 object categories present in the dataset. For MIMIC-CXR, we sample $t$ from a set of five disease categories (cardiomegaly, pneumothorax, atelectasis, pleural effusion, and edema) and $v$ from a set of five medical devices (pacemaker, chest tube, endotracheal tube, surgical clips, sternotomy wires).[2]

3. **Injecting the predefined systematic misalignment.** We insert the erroneous textual fact $t$ into text samples in the base vision-language dataset such that a strong association exists between text containing $t$ and images containing visual feature $v$. The strength of the association is controlled using Cramer's V scores. Each inserted fact $t$ is formatted as a sentence using diverse templates.

We repeat this procedure across the two possible options for the base dataset and a range of possible options for $t$ and $v$, yielding 420 evaluation settings encompassing a total of 1.7 million image-text pairs. Additional details are in Appendix B and C.

**Benchmark Evaluation:** We will use the notation $\{(\mathcal{D}_s, (t_s, v_s))\}_{s=1}^{420}$ to represent SYMBALBENCH, where the evaluation setting with index $s$ has an associated dataset $\mathcal{D}_s$ and ground-truth label $(t_s, v_s)$. We construct both reference-based and reference-free variants of SYMBALBENCH, which differ only with respect to whether $\mathcal{D}_s$ includes reference captions. At evaluation time, dataset $\mathcal{D}_s$ will be provided to method $\mathcal{M}$, which will output a prediction $(\hat{t}_s, \hat{v}_s)$. We count the prediction as accurate if the top-K predictions for $\hat{t}_s$ include $t_s$ and the top-K predictions for $\hat{v}_s$ include $v_s$. Here, we evaluate equivalence using LLM-as-a-Judge with Llama3.3-70B (Grattafiori et al., 2024). Overall performance on SYMBALBENCH is measured with Accuracy@K, computed as the percentage of the 420 settings in SYMBALBENCH where the prediction is accurate.

## 6. Results

We now evaluate SYMBAL on the systematic misalignment detection task. In Sections 6.1 and 6.2, we use SYMBAL-

---

[2]We define these options for $t$ and $v$ due to the fact that medical imaging models often learn spurious associations between medical devices and disease categories, as documented in prior work (e.g. (Oakden-Rayner et al., 2020)); thus, our predefined misalignments are highly plausible in real-world, model-generated reports.

BENCH to analyze the choice of embedding models, alignment scorers, and summarizers. In Section 6.3, we perform end-to-end evaluations of the best configuration of SYMBAL, comparing with baselines and performing fine-grained analyses. Finally, in Section 6.4, we extend beyond SYMBALBENCH to real-world settings.

### 6.1. SYMBAL Detects Erroneous Textual Facts

We first evaluate the role of various text embedding models, alignment scorers, and summarizers on the performance of Stage 1 of SYMBAL, which aims to predict the erroneous textual fact $\hat{t}_s$ given an input dataset $\mathcal{D}_s$ in SYMBALBENCH. We compute Accuracy@1 and Accuracy@5 by comparing $\hat{t}_s$ with $t_s$ across all 420 settings in SYMBALBENCH. Results are summarized in Table 1.

For the natural image datasets in SYMBALBENCH, Table 1 Upper demonstrates the performance of the top-four compositions, ranked by Accuracy@5 scores on the reference-free setting. Our results show that the best-performing variant of SYMBAL (shown in Row 1 of Table 1 Upper) achieves strong performance, correctly identifying the erroneous textual fact in 94.2% (Acc@5) of SYMBALBENCH datasets in the reference-free configuration and 82.8% (Acc@5) of SYMBALBENCH datasets in the reference-based configuration. Interestingly, we find that performance in reference-free settings is often substantially higher than performance in the reference-based setting, which is likely a result of the sparse information content often present in COCO reference captions. When considering the composition of SYMBAL, we note that the choice of the alignment scorer appears to be most important; the vision-language scorer substantially outperforms the text-only scorer with the same underlying model (Qwen2.5-72B). Given these results, we select the Qwen3-Embedding-8B text embedding model (Zhang et al., 2025), the vision-language alignment scorer with Qwen2.5-72B (Qwen et al., 2025), and the text-only summarizer with Qwen2.5-72B (Qwen et al., 2025) for all future SYMBAL evaluations on natural images.

For the medical image datasets in SYMBALBENCH, Table 1 Lower demonstrates the performance of the top-four compositions. Our results show that the best-performing variant of SYMBAL (shown in Row 1 of Table 1 Lower) correctly identifies the erroneous textual feature in 75.0% (Acc@5) of datasets in the reference-free configuration and 95.0% (Acc@5) of datasets in the reference-based configuration. In contrast to the natural image datasets, we find that the reference-free configuration is harder than the reference-based configuration, likely due to the complexity of medical image data; alignment scoring in this domain is challenging without access to reference text. We also note that a key advantage of SYMBAL is its ability to extend to specialized domains simply by interchanging constituent models with

*Table 1.* We evaluate various text embedding models, alignment scorers, and summarizers on the performance of SYMBAL Stage 1.

| | Text Embedding | Alignment Scorer | Summarizer | Reference-Free | | Reference-Based | |
|---|---|---|---|---|---|---|---|
| | | | | Acc@1 | Acc@5 | Acc@1 | Acc@5 |
| Natural | Qwen3-8B | Vision-Language (Qwen-72B) | Text-Only (Qwen-72B) | **92.8** | **94.2** | 80.8 | 82.8 |
| | OpenCLIP | Vision-Language (Qwen-72B) | Text-Only (Qwen-72B) | **92.8** | 93.9 | **86.1** | **87.8** |
| | Qwen3-8B | Text-Only (Qwen-72B) | Text-Only (Qwen-72B) | 82.8 | 85.0 | 81.9 | 83.9 |
| | OpenCLIP | Text-Only (Qwen-72B) | Text-Only (Qwen-72B) | 64.2 | 67.2 | 67.5 | 71.4 |
| Medical | XRayCLIP | Text-Only (MedGemma-27B) | Text-Only (MedGemma-27B) | **51.7** | **75.0** | 88.3 | 95.0 |
| | XRayCLIP | Text-Only (MedGemma-27B) | Text-Only (Qwen-72B) | **51.7** | 73.3 | **100.0** | **100.0** |
| | XRayCLIP | Text-Only (Qwen-72B) | Text-Only (MedGemma-27B) | 26.7 | 58.3 | 90.0 | 93.3 |
| | MedSigLIP | Text-Only (MedGemma-27B) | Text-Only (MedGemma-27B) | 30.0 | 53.3 | 83.3 | **100.0** |

*Table 2.* We evaluate various image embedding models, alignment scorers, and summarizers on the performance of SYMBAL Stage 2.

| | Image Embedding | Alignment Scorer | Summarizer | Reference-Free | | Reference-Based | |
|---|---|---|---|---|---|---|---|
| | | | | Acc@1 | Acc@5 | Acc@1 | Acc@5 |
| Natural | OpenCLIP | Vision-Language (Qwen-72B) | Text-Only (Qwen-72B) | **49.7** | **69.7** | 41.9 | 52.2 |
| | OpenCLIP | Embedding (OpenCLIP) | Vision-Language (Qwen-72B) | 48.1 | 63.9 | 42.5 | 55.6 |
| | OpenCLIP | Embedding (OpenCLIP) | Text-Only (Qwen-72B) | 47.8 | 62.8 | **43.9** | **55.8** |
| | OpenCLIP | Vision-Language (Qwen-72B) | Vision-Language (Qwen-72B) | 45.8 | 62.5 | 38.9 | 52.2 |
| Medical | XRayCLIP | Embedding (MedSigLIP) | Vision-Language (MedGemma-27B) | 11.7 | **36.7** | 28.3 | 53.3 |
| | MedSigLIP | Embedding (MedSigLIP) | Vision-Language (MedGemma-27B) | 11.7 | 31.7 | 25.0 | 46.7 |
| | OpenCLIP | Embedding (MedSigLIP) | Vision-Language (MedGemma-27B) | **13.3** | 28.3 | 20.0 | 46.7 |
| | MedSigLIP | Embedding (XRayCLIP) | Vision-Language (MedGemma-27B) | 10.0 | 28.3 | **33.3** | **60.0** |

domain-specific versions. Given these results, we select the XRayCLIP-ViT-L text embedding model (Chen et al., 2024c), the text-only alignment scorer with MedGemma-27B (Sellergren et al., 2025), and the text-only summarizer with MedGemma-27B (Sellergren et al., 2025) for all future SYMBAL evaluations on medical images.

### 6.2. SYMBAL Detects Associated Visual Features

We next evaluate the role of various image embedding models, alignment scorers, and summarizers on the performance of Stage 2 of SYMBAL. We hold the composition of Stage 1 constant using results from Section 6.1. We compute Accuracy@1 and Accuracy@5 by comparing $\hat{v}_s$ with $v_s$ across all 420 settings in SYMBALBENCH. Results are summarized in Table 2.

For the natural image datasets in SYMBALBENCH, Table 2 Upper demonstrates the performance of the top-four compositions, ranked by Accuracy@5 scores on the reference-free setting. Our results show that the best-performing variant of SYMBAL (shown in Row 1 of Table 2 Upper) correctly identifies the visual feature in 69.7% (Acc@5) of datasets in the reference-free configuration and 52.2% (Acc@5) of datasets in the reference-based configuration. We observe that performance values in Table 2 are lower than Table 1, suggesting that identifying visual features that systematically occur with textual errors is substantially more challenging than identifying the textual error itself. We also

observe that the best-performing variant of SYMBAL utilizes the same alignment scorer and summarizer as in Stage 1. Given these results, we select the OpenCLIP-ViT-H image embedding model (Ilharco et al., 2021), vision-language alignment scorer with Qwen2.5-72B (Qwen et al., 2025), and text-only summarizer with Qwen2.5-72B (Qwen et al., 2025) for all future SYMBAL evaluations on natural images.

For the medical image datasets in SYMBALBENCH, Table 2 Lower demonstrates the performance of the top-four compositions, ranked by Accuracy@5 scores on the reference-free setting. Our results show that the best-performing variant of SYMBAL (shown in Row 1 of Table 2 Lower) correctly identifies the visual feature in 36.7% (Acc@5) of datasets in the reference-free configuration and 53.3% (Acc@5) of datasets in the reference-based configuration. Our results suggest that identifying visual features in the medical domain is a particularly challenging task in both reference-free and reference-based settings, and consequently, the optimal composition of alignment scorers and summarizers differs markedly from those identified in Stage 1. Given these results, we select the XRayCLIP-ViT-L image embedding model (Chen et al., 2024c), embedding alignment scorer with MedSigLIP (Sellergren et al., 2025), and vision-language summarizer with MedGemma-27B (Sellergren et al., 2025) for all future evaluations on medical images.

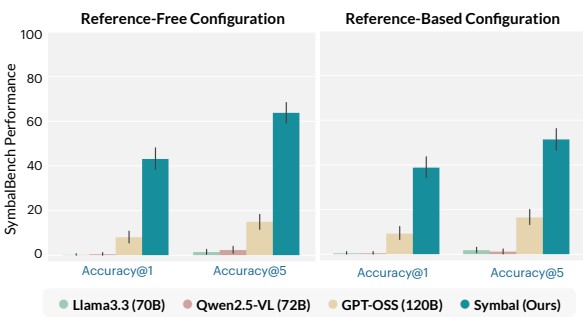

*Figure 3.* SYMBAL demonstrates strong end-to-end performance on SYMBALBENCH, substantially outperforming baselines.

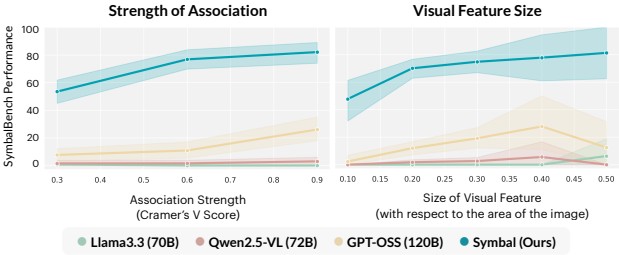

*Figure 4.* We report performance on SYMBALBENCH (reference-free) stratified across association strengths and visual feature sizes. This analysis focuses on natural image settings in SYMBALBENCH.

## 6.3. SYMBAL Shows Strong End-to-End Performance

Given an optimal composition of SYMBAL, we now perform end-to-end analyses across SYMBALBENCH. Since our study proposes a novel task, there are no existing baselines for comparison. As a result, we compare the structured, dual-stage approach of SYMBAL to a single-stage, direct-prompting method where each dataset $\mathcal{D}_s$ is directly provided to an off-the-shelf LLM in the form of a text prompt; the LLM is then instructed to output the erroneous textual fact and the associated visual feature. Three state-of-the-art LLMs are considered (i.e. Llama3.3 70B, Qwen2.5-VL 72B, and GPT-OSS 120B), selected to ensure a fair comparison with SYMBAL due to comparable parameter counts. As the token length of the direct prompts far surpasses the context window of these LLMs, we use only a sample of each dataset, ensuring that the final inference procedure requires no more compute resources than SYMBAL.

In Figure 3, we measure the extent to which SYMBAL can accurately predict *both* the textual fact $\hat{t}_s$ and the visual feature $\hat{v}_s$ across both the reference-free and reference-based variants of SYMBALBENCH. Results show that the systematic misalignment detection task is highly challenging in both experimental settings, with several baselines generating few correct predictions. SYMBAL successfully identifies the systematic misalignment in up to 63.8% of datasets in SYMBALBENCH, with the highest performance observed in the reference-free setting (Accuracy@5). SYMBAL outperforms the closest baseline (GPT-OSS 120B) across all

experimental settings, with GPT-OSS 120B correctly identifying the misalignment in only 17.1% of SYMBALBENCH datasets in the best case. These results demonstrate that the structured, dual-stage approach utilized by SYMBAL provides substantial performance benefits over single-stage, direct prompting baselines.

In Figure 4, we provide a stratified breakdown of SYMBAL performance. SYMBAL outperforms baselines across highly-challenging subsets of SYMBALBENCH where (1) the strength of the systematic misalignment is weak (i.e. weak association between the textual error and visual feature as measured by Cramer's V scores) and (2) visual features are small in size.

Extended results and ablations are provided in Appendix Section D.

## 6.4. SYMBAL Extends to Real-World Settings

In this section, we further demonstrate the utility of SYMBAL by supplementing our evaluations on SYMBALBENCH with additional quantitative and qualitative analyses in real-world settings. Our results show that (1) SYMBAL can accurately surface systematic misalignments in captions generated by off-the-shelf MLLMs and (2) SYMBAL is a powerful tool for auditing vision-language datasets.

**SYMBAL can accurately surface systematic misalignments in captions generated by off-the-shelf MLLMs.** First, we use SYMBAL to analyze captions generated by four real-world off-the-shelf MLLMs: Llava1.5-7B (Liu et al., 2024), Llava1.5-13B (Liu et al., 2024), AyaVision-8B (Dash et al., 2025), and LlavaOneVision-7B (Li et al., 2025). We utilize each model to generate captions for the COCO dataset (2017 val split); we then apply SYMBAL (reference-free) to predict systematic misalignments $(\hat{t}, \hat{v})$.

As discussed in Section 5, evaluating predictions in real-world settings is highly challenging since ground-truth systematic misalignments are unknown. Here, in order to address this issue, we validate identified systematic misalignments in two ways. First, we *qualitatively* validate the existence of SYMBAL-identified systematic misalignments with visual analysis. Second, we *quantitatively* validate whether a link between erroneous fact $\hat{t}$ and visual feature $\hat{v}$ truly exists; to this end, we measure whether model-generated captions are indeed more likely to include erroneous references to $\hat{t}$ when $\hat{v}$ is present compared to when $\hat{v}$ is absent. In order to perform this evaluation, we use a state-of-the-art open-set object detector (Minderer et al., 2023) to annotate the presence of $\hat{v}$ in each image, and we use our top-performing alignment scorer (vision-language scorer with Qwen-72B) to annotate erroneous references to $\hat{t}$ in each caption. In Appendix E, we demonstrate that automated annotations align closely with human judgments.

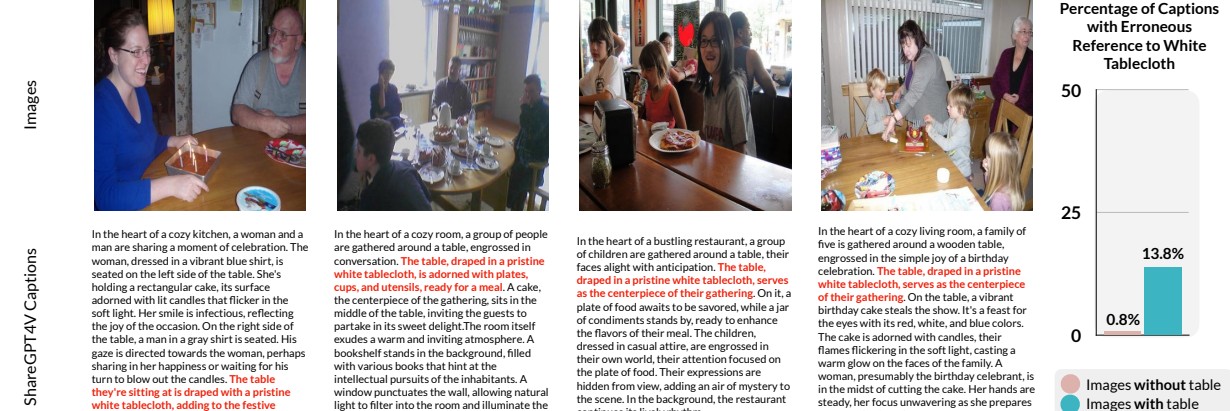

Figure 5. SYMBAL discovers systematic misalignments in ShareGPT4V, an off-the-shelf dataset with model-generated captions.

SYMBAL identifies several systematic misalignments. In captions generated by Llava1.5-7B, SYMBAL detects that erroneous references to a `handbag` or a `handbag on the ground` ($\hat{t}$) in captions are often systematically associated with the presence of a `bus` ($\hat{v}$) in a scene, as shown in Figure 9 [Row 2]. Quantitatively, our analysis finds that erroneous references to a `handbag` in model-generated captions are indeed 3.1 times more likely when a `bus` is present in the image compared to when a `bus` is absent, validating the SYMBAL prediction. In captions generated by LlavaOneVision-7B, SYMBAL detects that erroneous references to `text` ($\hat{t}$) in captions are often systematically associated with the presence of a `sign` ($\hat{v}$) in a scene, as shown in Figure 10 [Row 2]. This finding suggests that LlavaOneVision-7B struggles with OCR capabilities, where the presence of text-based signage in an image is likely to result in errors in the generated caption. Quantitatively, our analysis finds that erroneous references to `text` in model-generated captions are indeed 4.6 times more likely when a `sign` is present in the image compared to when a `sign` is absent, validating the SYMBAL prediction. Additional examples can be found in Appendix E.

**SYMBAL is a powerful tool for auditing open-source vision-language datasets.** Second, we use SYMBAL to analyze ShareGPT4V, an open-source image dataset with MLLM-generated captions commonly used as a pretraining dataset for vision-language models (Chen et al., 2024b). We sample a subset of 10k image-caption pairs from the ShareGPT4V dataset, and we then apply SYMBAL (reference-free) to predict systematic misalignments ($\hat{t}$, $\hat{v}$). Here, SYMBAL detects that erroneous references to a `white tablecloth` ($\hat{t}$) in captions are often systematically associated with the presence of a `table`, `cake`, and/or `people` ($\hat{v}$) in the scene, as shown in Figure 5. Quantitatively, our analysis finds that erroneous references to a `white tablecloth` in model-generated captions are indeed 17.2 times more likely when a `table` is present in the image compared to when a `table` is absent, validating the SYMBAL prediction. Additional examples are provided in Appendix E.

As large-scale datasets like ShareGPT4V become increasingly prevalent, it becomes critical for users to be aware of potential systematic misalignments, as these errors can propagate to trained models. Specifically, if a dataset contains a systematic misalignment between erroneous textual fact $\hat{t}$ and visual feature $\hat{v}$, models trained on the dataset are likely to learn spurious correlations between $\hat{t}$ and $\hat{v}$, leading to prediction errors at test-time (Varma et al., 2024). SYMBAL can aid users with understanding limitations of datasets with MLLM-generated captions as well as assist model developers with improving performance of MLLMs.

## 7. Discussion

In this work, we introduce the systematic misalignment detection task, which aims to identify textual errors in MLLM-generated captions that are systematically associated with visual features. We hope that our novel task, method SYMBAL, and benchmark SYMBALBENCH can help users audit MLLM-generated captions and identify critical failure modes, even without access to the underlying MLLM.

## Impact Statement

The goal of our work is to improve transparency into a critical class of captioning errors in image-text datasets. As datasets with model-generated captions gain in popularity and become widely adopted into training datasets for the next generation of multimodal foundation models, it becomes critical to audit data and understand potential quality issues before use. We hope that our novel task, benchmark, and method can help make progress towards this goal, particularly in safety-critical domains like medicine.

## Acknowledgments

MV is supported by graduate fellowship awards from the Knight-Hennessy Scholars program at Stanford University, the Quad program, and the United States Department of Defense (NDSEG). AC is supported by NIH grants R01 HL167974, R01HL169345, R01 AR077604, R01 EB002524, R01 AR079431, P41 EB027060, AY2 AX000045, and 1AYS AX0000024-01; ARPA-H grants AY2AX000045 and 1AYSAX0000024-01; and NIH contracts 75N92020C00008 and 75N92020C00021. AC has provided consulting services to Patient Square Capital, Chondrometrics GmbH, and Elucid Bioimaging; is co-founder of Cognita; has equity interest in Cognita, Subtle Medical, LVIS Corp, Brain Key. CL is supported by NIH grants R01 HL155410, R01 HL157235, by AHRQ grant R18HS026886, and by the Gordon and Betty Moore Foundation. CL is also supported by the Medical Imaging and Data Resource Center (MIDRC), which is funded by the National Institute of Biomedical Imaging and Bioengineering (NIBIB) under contract 75N92020C00021 and through the Advanced Research Projects Agency for Health (ARPA-H).

This research was funded, in part, by the Advanced Research Projects Agency for Health (ARPA-H). The views and conclusions contained in this document are those of the authors and should not be interpreted as representing the official policies, either expressed or implied, of the U.S. Government.

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

# Appendix

## Contents

## A. Implementation Details for SYMBAL

SYMBAL decomposes the systematic misalignment detection task into two stages; here, we provide extended implementation details for each of these stages.

### A.1. Implementation Details for SYMBAL Stage 1

**Subtask 1: Grouping semantically-similar facts.** We express each text sample $T_i$ as a collection of textual facts $T_i = \{t_1^i, t_2^i, ..., t_{n_i}^i\}$ by splitting captions at the sentence-level. We opt to use sentence-level splitting in this work because each sentence in a long-form caption typically captures a semantically-meaningful, self-contained fact. Sentence-level splitting has been utilized in prior literature (e.g. (Zhang et al., 2022)). We note here that there may be settings where this strategy is sub-optimal, such as when a sentence does not represent a self-contained fact and instead relies on previous context. In such cases, users of SYMBAL can easily adjust this design choice by modifying the definition of "textual fact" to cover relevant context.

After aggregating all textual facts in $\mathcal{D}$ forming the set $\bigcup_{i=1}^{N} T_i$, we encode each fact using a text embedding model. For natural image datasets in SYMBALBENCH derived from COCO, we consider two options for text embedding models: OpenCLIP-ViT-H-14-quickgelu (Ilharco et al., 2021) and Qwen3-Embedding-8B (Zhang et al., 2025). For medical image datasets in SYMBALBENCH derived from MIMIC-CXR, we consider three options for text embedding models: OpenCLIP-ViT-H-14-quickgelu (Ilharco et al., 2021), XRayCLIP-ViT-L (Chen et al., 2024c), and MedSigLIP (Sellergren et al., 2025). Of these, XrayCLIP-ViT-L and MedSigLIP are trained on radiology datasets. Embeddings are then clustered using spherical K-Means (implemented in Faiss (Johnson et al., 2019b)), where we sweep across a range of potential cluster numbers and select the optimal number of clusters using Silhouette distance; this approach is motivated by prior work (Sohoni et al., 2020; Varma et al., 2025).

**Subtask 2: Scoring groups by degree of misalignment.** We score each cluster by computing the average degree of alignment between constituent textual facts and paired images. We consider three possible scoring mechanisms, explained in detail below:

- *Embedding scorer:* Given a textual fact and its paired image, the embedding scorer utilizes an off-the-shelf vision-language model to compute embeddings for the text and image modalities. Alignment is measured by computing cosine similarity. This method is motivated by metrics like CLIPScore (Hessel et al., 2021), which have shown strong correlation with human judgments when measuring caption quality. For natural image datasets in SYMBALBENCH derived from COCO, we implement the embedding scorer with OpenCLIP-ViT-H-14-quickgelu (Ilharco et al., 2021) as the vision-language model. For medical image datasets in SYMBALBENCH derived from MIMIC-CXR, we consider three options for the embedding scorer: OpenCLIP-ViT-H-14-quickgelu (Ilharco et al., 2021), XRayCLIP-ViT-L (Chen et al., 2024c), and MedSigLIP (Sellergren et al., 2025). We note here that we do not alter the embedding scorer for reference-based settings; reference captions $R_i$ in our benchmark often have substantially more information than the single textual fact $t_k^i \in T_i$, and this information imbalance is challenging to capture with embedding scorers.

- *Text-only scorer:* Given a textual fact and its paired image, the text-only scorer first generates a caption for the image and then prompts an LLM to determine if the textual fact is accurate with respect to the caption. For natural image datasets in SYMBALBENCH derived from COCO, we implement the text-only scorer using Llama-3.2-11B-Vision-Instruct

(Grattafiori et al., 2024) to generate captions and Qwen2.5-VL-72B-Instruct (Qwen et al., 2025) to perform scoring. For medical image datasets in SYMBALBENCH derived from MIMIC-CXR, we implement the text-only scorer using Maira-2 (Bannur et al., 2024) to generate captions and Qwen2.5-VL-72B-Instruct (Qwen et al., 2025) or MedGemma-27B (Sellergren et al., 2025) to perform scoring. In the reference-based setting, we use the ground-truth caption $R_i$ rather than generating captions. We use the following input prompt in order to perform scoring:

---

**Text-Only Scorer Input Prompt**

You are provided with two image captions below, denoted as [A] and [B].
[A]: <generated image caption or ground-truth reference caption>
[B]: <candidate textual fact>
Assume that [A] is the ground-truth caption. Is the content of [B] factually accurate with respect to [A]?
Rules:
1. [B] may omit details from [A]; omission is acceptable.
2. If [B] introduces any incorrect or contradictory detail, it is inaccurate.
Please output your answer as a single digit, where 1 indicates that [B] is accurate and 0 indicates that [B] is not accurate. Do not provide anything other than the digit in your response.

---

• *Vision-language scorer:* Given a textual fact and its paired image, the vision-language scorer provides an MLLM with both the image and the textual fact as input; the MLLM is then tasked with determining if the textual fact is accurate. For natural image datasets in SYMBALBENCH derived from COCO, we utilize Qwen2.5-VL-72B-Instruct (Qwen et al., 2025) as the MLLM. For medical image datasets in SYMBALBENCH derived from MIMIC-CXR, we utilize MedGemma-27B (Sellergren et al., 2025) as the MLLM. We use the following input prompt in the reference-free setting:

---

**Vision-Language Scorer Input Prompt (Reference-Free)**

<image>
You are given an image. Below, a caption for the image is provided:
Caption: <candidate textual fact>
Is the caption accurate with respect to the image? Please output your answer as a single digit, where 1 indicates that the caption is accurate and 0 indicates that the caption is not accurate. Do not provide anything other than the digit in your response.

---

In the reference-based setting, we additionally provide the ground-truth reference caption to the MLLM. We use the following prompt in the reference-based setting:

---

**Vision-Language Scorer Input Prompt (Reference-Based)**

<image>
You are provided an image as well as two image captions below, denoted as [A] and [B].
[A]: <ground-truth reference caption>
[B]: <candidate textual fact>
Assume that [A] is the ground-truth caption. Is the content of [B] accurate with respect to the image? Please output your answer as a single digit, where 1 indicates that the caption is accurate and 0 indicates that the caption is not accurate. Do not provide anything other than the digit in your response.

---

**Subtask 3: Summarizing the top-ranked group.** We consider the following summarization mechanism for identifying the unifying concept shared by textual facts in $C_{text}$.

• *Text-only summarizer:* The text-only summarizer provides an LLM with textual facts in $C_{text}$; the LLM is then tasked with identifying the unifying concept. For natural image datasets in SYMBALBENCH derived from COCO, we use Qwen2.5-VL-72B-Instruct (Qwen et al., 2025) as the LLM. For medical image datasets in SYMBALBENCH derived from MIMIC-CXR, we consider both Qwen2.5-VL-72B-Instruct (Qwen et al., 2025) and MedGemma-27B (Sellergren et al., 2025) as the LLM.

We use the following input prompt. Then, given the output, we prompt the same LLM to select the most frequently identified feature (or the top-k most frequently identified features) as output.

---

**Text-Only Summarizer Input Prompt**

Consider this image caption: "<candidate textual fact>"
Identify the visual features that are present in the image.
Output your answer in the following format:
Answer: comma-separated list

Rules:
1. Each feature should be described concisely in a single phrase.
2. Each feature must be directly visible in the image.
3. Do NOT include any text outside the identified features.
4. Do NOT explain your reasoning.
5. If no features are present, output an empty list of the form: "Answer: "

---

### A.2. Implementation Details for SYMBAL Stage 2

**Subtask 1: Grouping semantically-similar images.** For natural image datasets in SYMBALBENCH derived from COCO, we consider two options for image embedding models: OpenCLIP-ViT-H-14-quickgelu (Ilharco et al., 2021) and DINOv2-ViT-L-14 (Oquab et al., 2024). For medical image datasets in SYMBALBENCH derived from MIMIC-CXR, we consider three options for image embedding models: OpenCLIP-ViT-H-14-quickgelu (Ilharco et al., 2021), XRayCLIP-ViT-L (Chen et al., 2024c), and MedSigLIP (Sellergren et al., 2025). Similar to Stage 1, embeddings are clustered using spherical K-Means, where we sweep across a range of cluster numbers and select the optimal number using Silhouette distance.

**Subtask 2: Scoring groups by degree of misalignment.** We score each cluster by computing the mean degree of misalignment between images and paired textual facts in $C_{text}$. We consider the same scoring mechanisms as in Stage 1.

**Subtask 3: Summarizing the top-ranked group.** We consider two summarization mechanisms for identifying the unifying concept shared by images in $C_{image}$, described in detail below.

- *Text-only summarizer:* The text-only summarizer generates a caption for each image in $C_{image}$; then, an LLM is tasked with identifying the unifying concept. For natural image datasets in SYMBALBENCH derived from COCO, captions are generated using Llama-3.2-11B-Vision-Instruct (Grattafiori et al., 2024). For medical image datasets in SYMBALBENCH, captions are generated using MAIRA-2 (Bannur et al., 2024). In reference-based settings, we use the ground-truth reference captions rather than generating. We use the same prompts and models as Stage 1, Subtask 3.

- *Vision-language summarizer:* The vision-language summarizer provides an MLLM with images in $C_{image}$; then, the MLLM is prompted to identify the unifying concept. For natural image datasets in SYMBALBENCH derived from COCO, we use Qwen2.5-VL-72B-Instruct (Qwen et al., 2025) as the MLLM. For medical image datasets in SYMBALBENCH derived from MIMIC-CXR, we use MedGemma-27B (Sellergren et al., 2025) as the MLLM. For reference-based settings, we also provide the ground-truth reference caption to the MLLM.

We use the following input prompt. Then, given the outputs, we prompt the same MLLM to select the most frequently identified feature (or the top-k most frequently identified features) as output.

---

**Vision-Language Summarizer Input Prompt**

<image>
Consider this image.

Identify the visual features that are present in the image.
Output your answer in the following format:

---

Answer: comma-separated list
Rules:
1. Each feature should be described concisely in a single phrase.
2. Each feature must be directly visible in the image.
3. Do NOT include any text outside the identified features.
4. Do NOT explain your reasoning.
5. If no features are present, output an empty list of the form: "Answer: "
6. Include a maximum of ten features.

### A.3. Extension to Multiple Systematic Misalignments

Real-world datasets are likely to include multiple systematic misalignments, and SYMBAL can be trivially extended to such settings as follows. Stage 1 of SYMBAL involves predicting the erroneous textual fact $\hat{t}$; here, rather than summarizing the single top ranked group of facts into a unifying concept, we can simply consider the top-k ranked groups instead. This will result in multiple predicted textual facts $\hat{t}^{(1)}, \hat{t}^{(2)}, ...\hat{t}^{(k)}$, each representing a distinct recurring textual error in the dataset. Stage 2 of SYMBAL can then be implemented as described in Section 4.2, taking into account each predicted textual fact; this will result in associated visual features $\hat{v}^{(1)}, \hat{v}^{(2)}, ...\hat{v}^{(k)}$. Ultimately, at the conclusion of this procedure, SYMBAL will predict multiple systematic misalignments $(\hat{t}^{(i)}, \hat{v}^{(i)})$ where $i$ ranges from 1 to $k$. In Figure 9, we empirically show that Symbal can accurately detect multiple real-world systematic misalignments in captions generated by Llava1.5-7B.

## B. Implementation Details for SYMBALBENCH

SYMBALBENCH is comprised of 420 evaluation settings, where 360 settings include natural image datasets derived from COCO and 60 settings include medical image datasets derived from MIMIC-CXR. Below, we provide extended implementation details for the natural image settings:

1. **Obtaining a base dataset.** The base vision-language datasets in the natural image domain are derived from COCO (2017 val split), which consists of photographs depicting common objects (e.g. animals, food, furniture, etc.) in natural settings. Images are paired with object-level annotations as well as five human-written captions, with each caption typically consisting of a single sentence or phrase describing salient features in the image. In order to ensure that objects are clearly visible in the image, we exclude annotations for all tiny objects, defined as objects that take up less than 5% of the area of the image. After filtering out images with no remaining object-level annotations, we are left with a base dataset consisting of 4349 images and associated captions. We then compose a new two-sentence caption for each image by randomly sampling two captions from the provided list of five captions.

2. **Predefining a systematic misalignment.** We then predefine a systematic misalignment consisting of a textual fact $t$ and the associated visual feature $v$. We sample $v$ from the set of 80 object categories present in the dataset. Then, we sample $t$ from the set of 80 object categories (such that $t \neq v$) utilizing three possible sampling strategies: (1) *random*, where $t$ is sampled randomly, (2) *popular*, where $t$ is sampled from the list of the top-ten most popular objects in the COCO training set, and (3) *adversarial*, where $t$ is the object that most commonly co-occurs with $v$ in the COCO training set. These sampling strategies are motivated by prior work (Li et al., 2023) and are meant to capture a range of possible error patterns that may emerge in real-world MLLM-generated captions.

3. **Injecting the predefined systematic misalignment.** We insert the erroneous textual fact $t$ into captions in the base dataset, ensuring that an association exists between text containing $t$ and images containing visual feature $v$; this procedure ensures that the misalignment is *systematic*. Importantly, we ensure that feature $t$ is not already in the image-caption pair prior to injection. We consider three levels of association, as measured by Cramer's V: low association (Cramer's V = 0.3), moderate association (Cramer's V = 0.6), and high association (Cramer's V = 0.9). In order to format textual fact $t$ into a sentence, we generate 50 templates using GPT-4o (OpenAI et al., 2024), select a template at random, and insert $t$.

We repeat this injection procedure for all possible choices of $t$ and $v$ in order to obtain 360 evaluation settings, each consisting of an image-caption dataset and paired annotation $(t,v)$.

Below, we provide extended implementation details for the medical image settings:

1. **Obtaining a base dataset.** The base vision-language datasets in the medical image domain are derived from MIMIC-CXR (test split), which consists of chest X-rays and associated radiologist reports collected at Beth Israel Deaconess Medical Center. We preprocess the dataset by (1) removing all images with non-frontal imaging views, (2) removing all images with missing "Impressions" sections in the paired report, and (3) removing all sentences in reports without "present" disease or anatomy entities, as identified by an off-the-shelf medical entity annotation tool (Delbrouck et al., 2024). After preprocessing, we are left with a base dataset consisting of 2233 images, each paired with the "Impressions" section of the corresponding report.

2. **Predefining a systematic misalignment.** We sample $t$ from a set of five disease categories selected from the commonly-used CheXpert annotation list (Irvin et al., 2019): cardiomegaly, pneumothorax, atelectasis, pleural effusion, and edema. We sample $v$ from a set of five medical devices: pacemaker, chest tube, endotracheal tube, surgical clips, sternotomy wires. We select these options for $t$ and $v$ since medical devices often co-occur with diseases, yet there is no deterministic, universal link. Models often learn spurious associations between devices and diseases as documented in prior work (Oakden-Rayner et al., 2020), meaning that such errors are highly plausible in MLLM-generated reports.

3. **Injecting the predefined systematic misalignment.** We insert the erroneous textual fact $t$ into reports in the base dataset, using Cramer's V to control the level of association with visual feature $v$. We use a combination of physician annotations, automated annotations from the CheXpert labeler (Irvin et al., 2019), and automated annotations from RadGraph-XL (Delbrouck et al., 2024) in order to identify whether or not $t$ and $v$ are present in the image-report pair prior to injection. In order to format textual fact $t$ into a sentence, we identify the 50 most frequently occurring sentences in the MIMIC-CXR training set that discuss the presence of $t$ and select a sentence from this list at random.

We repeat this injection procedure for all possible choices of $t$ and $v$ in order to obtain 60 evaluation settings, each consisting of an image-caption dataset and paired annotation ($t$,$v$).

In reference-based settings, we also include a ground-truth caption $R_i$ along with each image-text pair $(V_i, T_i) \in \mathcal{D}$. For natural image datasets derived from COCO, $R_i$ takes the form of a three-sentence caption combining the three human-written captions not originally selected as part of $T_i$. For medical image datasets derived from MIMIC-CXR, $R_i$ takes the form of the "Findings" and "Impressions" sections of the original physician-written radiology report. We emphasize that $T_i$ may contain errors as a result of the error-injection procedure detailed above; however, $R_i$ is always accurate.

We determine if predictions are equivalent to the ground-truth by leveraging LLM-as-a-Judge. We use Llama3.3-70B in all experiments as the LLM, leveraging the `ollama` implementation with default parameters. The input prompt is:

> **LLM-as-a-Judge Evaluation Prompt**
>
> You are given two short text phrases.
> Model response: <predicted textual error or predicted visual feature>
> Ground truth: <ground-truth textual error or ground-truth visual feature>
>
> Your task is to determine if both phrases refer to the same visual feature. Please output 1 if both the model response and the correct answer refer to the same feature or 0 if the model response and the correct answer do not refer to the same feature. Do not provide anything other than the number in your response.

## C. SYMBALBENCH Descriptive Statistics

In this section, we provide descriptive statistics summarizing the composition of SYMBALBENCH. SYMBALBENCH includes 420 settings covering two domains (with 360 natural image settings and 60 medical image settings). In Table 3, we provide a list of all ground-truth systematic misalignments $(t, v)$ included in SYMBALBENCH.

In Figure 6, we summarize SYMBALBENCH with histograms detailing (1) the size of each dataset, (2) the strength of the injected systematic misalignment in each dataset as measured with Cramer's V, (3) the proportion of image-text pairs in each dataset containing the injected textual error $t$, and (4) the proportion of image-text pairs in each dataset containing the visual feature $v$. In Figure 7, we provide additional descriptive statistics on the natural image subset of SYMBALBENCH consisting of datasets derived from COCO; here, we provide histograms detailing (1) the mean size of the visual feature in each dataset (measured as the proportion of the total image area) and (2) the category of systematic misalignment (random, popular, or adversarial) as discussed in Appendix Section B.

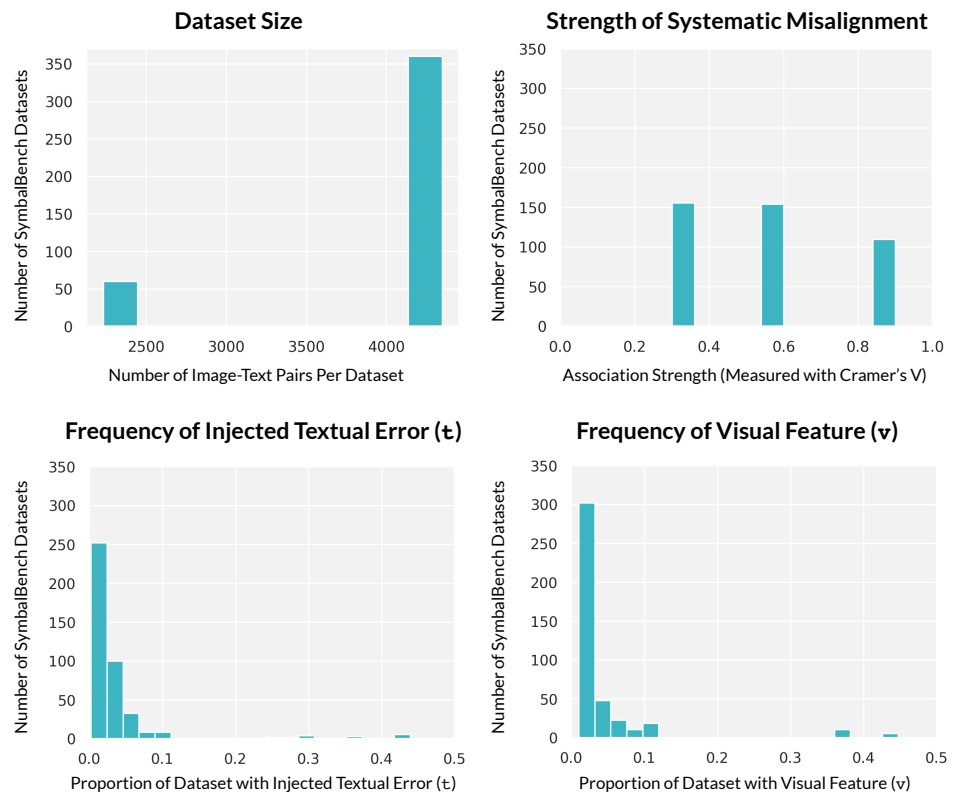

*Figure 6.* Here, we provide histograms summarizing the composition of datasets included in SYMBALBENCH.

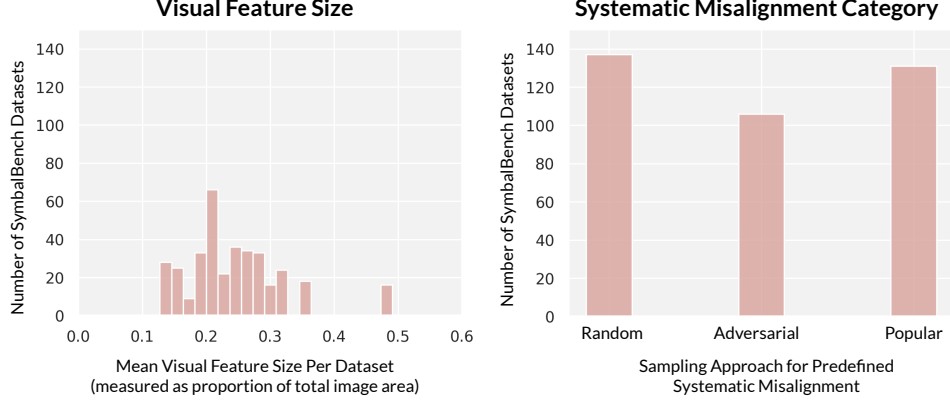

*Figure 7.* We provide additional descriptive statistics summarizing the composition of the 360 natural image datasets in SYMBALBENCH. We note here that if multiple sampling strategies yield the same predefined systematic misalignment, more than one category will be assigned to the same dataset; thus, the total count for the systematic misalignment category histogram may exceed 360.

*Table 3.* Here, we provide a list of all ground-truth systematic misalignments $(t, v)$ included in SYMBALBENCH.

| Erroneous Textual Fact $t$ | Visual Feature $v$ | Erroneous Textual Fact $t$ | Visual Feature $v$ | Erroneous Textual Fact $t$ | Visual Feature $v$ |
|---|---|---|---|---|---|
| surfboard | airplane | person | airplane | bottle | airplane |
| person | banana | chair | banana | car | banana |
| kite | bed | person | bed | chair | bed |
| person | bench | handbag | bench | oven | bench |
| hot dog | bicycle | person | bicycle | truck | bicycle |
| person | bird | wine glass | bird | book | bird |
| truck | boat | person | boat | bicycle | boat |
| toilet | book | cup | book | person | book |
| pizza | bottle | person | bottle | elephant | bowl |
| car | bowl | dining table | bowl | cat | broccoli |
| dining table | broccoli | car | broccoli | handbag | bus |
| frisbee | bus | person | bus | bicycle | cake |
| dining table | cake | chair | cake | fork | car |
| person | car | car | cat | umbrella | cat |
| person | cat | airplane | chair | person | chair |
| car | chair | bottle | couch | baseball glove | couch |
| person | couch | person | cow | cake | cow |
| bowl | cow | person | cup | bottle | cup |
| microwave | cup | book | dining table | apple | dining table |
| person | dining table | chair | dog | person | dog |
| laptop | dog | boat | elephant | person | elephant |
| bowl | elephant | dining table | fire hydrant | car | fire hydrant |
| airplane | fire hydrant | sandwich | fork | dining table | fork |
| car | fork | cup | giraffe | umbrella | giraffe |
| person | giraffe | cup | horse | person | horse |
| banana | horse | zebra | keyboard | truck | keyboard |
| mouse | keyboard | person | laptop | bottle | laptop |
| hair drier | motorcycle | book | motorcycle | person | motorcycle |
| giraffe | oven | sink | oven | cup | oven |
| laptop | person | car | person | dining table | pizza |
| person | pizza | cell phone | pizza | airplane | potted plant |
| person | potted plant | book | potted plant | dining table | refrigerator |
| microwave | refrigerator | oven | refrigerator | stop sign | sandwich |
| dining table | sandwich | dining table | sheep | person | sheep |
| orange | sheep | cat | sink | car | sink |
| bottle | sink | fork | suitcase | person | suitcase |
| bowl | surfboard | airplane | surfboard | person | surfboard |
| carrot | teddy bear | bowl | teddy bear | person | teddy bear |
| bottle | toilet | car | toilet | sink | toilet |
| cup | train | person | train | truck | train |
| dining table | truck | refrigerator | truck | person | truck |
| spoon | tv | chair | tv | car | tv |
| baseball bat | umbrella | person | umbrella | tv | zebra |
| giraffe | zebra | book | zebra | cardiomegaly | surgical clips |
| edema | chest tube | pleural effusion | chest tube | pneumothorax | chest tube |
| atelectasis | chest tube | cardiomegaly | chest tube | edema | endotracheal tube |
| pleural effusion | endotracheal tube | atelectasis | endotracheal tube | pneumothorax | endotracheal tube |
| cardiomegaly | endotracheal tube | edema | pacemaker | pleural effusion | pacemaker |
| pneumothorax | pacemaker | atelectasis | pacemaker | cardiomegaly | pacemaker |
| atelectasis | sternotomy wires | pneumothorax | sternotomy wires | cardiomegaly | sternotomy wires |
| edema | sternotomy wires | pleural effusion | sternotomy wires | edema | surgical clips |
| pleural effusion | surgical clips | atelectasis | surgical clips | pneumothorax | surgical clips |

# D. Extended Results

In Table 4, we provide an extended version of Table 1, extending to the top-ten compositions. Note that Table 4 excludes compositions consisting of an embedding-based alignment scorer and text-only summarizer, as this combination does not make use of reference captions in the reference-based setting.

In Table 5, we provide an extended version of Table 2, extending to the top-ten compositions. Again, Table 5 only includes compositions that can support both SYMBALBENCH variants.

In Table 6, we provide a tabular version of Figure 3 stratified by domain.

In Figure 8, we extend Figure 4 by providing a breakdown of SYMBAL performance across various categories of systematic misalignments in the natural image subset of SYMBALBENCH.

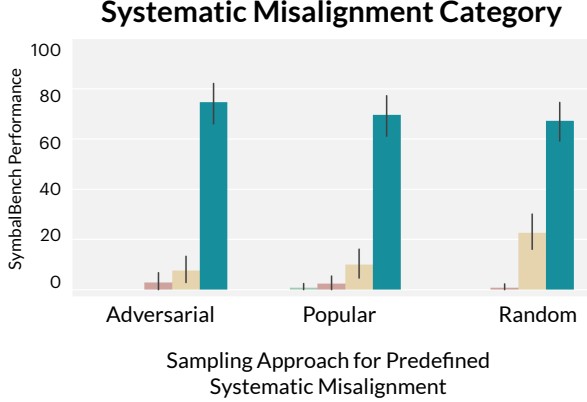

*Figure 8.* We provide a breakdown of SYMBAL performance across various categories of systematic misalignments in the natural image subset of SYMBALBENCH.

We use the following input prompt for our direct-prompting baselines:

---

**Direct-Prompting Baseline Input Prompt**

You are provided with a dataset, where each sample consists of the following two components:

Reference caption: A ground-truth caption describing the content of an image
Model-generated caption: A caption generated by an AI model

The model-generated captions may have systematic errors, where a recurring textual error is closely associated with the presence of a specific visual feature in the paired image. Your task is to identify the recurring textual error and the associated visual feature.

Output your answer in the following format, where each comma-separated list consists of your top-five predictions in order:
Textual Error: comma-separated list
Visual Feature: comma-separated list

Rules:
1. Each visual feature must be directly visible in the image.
2. Do NOT include any text outside of the answer.
3. Do NOT explain your reasoning.

Dataset: <samples from dataset with images expressed in text-form>

---

*Table 4.* We evaluate various text embedding models, alignment scorers, and summarizers on the performance of Stage 1 of SYMBAL.

| | Text Embedding | Alignment Scorer | Summarizer | Reference-Free | | Reference-Based | |
|---|---|---|---|---|---|---|---|
| | | | | Acc@1 | Acc@5 | Acc@1 | Acc@5 |
| Natural | Qwen3-8B | Vision-Language (Qwen-72B) | Text-Only (Qwen-72B) | **92.8** | **94.2** | 80.8 | 82.8 |
| | OpenCLIP | Vision-Language (Qwen-72B) | Text-Only (Qwen-72B) | **92.8** | 93.9 | **86.1** | **87.8** |
| | Qwen3-8B | Text-Only (Qwen-72B) | Text-Only (Qwen-72B) | 82.8 | 85.0 | 81.9 | 83.9 |
| | OpenCLIP | Text-Only (Qwen-72B) | Text-Only (Qwen-72B) | 64.2 | 67.2 | 67.5 | 71.4 |
| Medical | XRayCLIP | Text-Only (MedGemma-27B) | Text-Only (MedGemma-27B) | **51.7** | **75.0** | 88.3 | 95.0 |
| | XRayCLIP | Text-Only (MedGemma-27B) | Text-Only (Qwen-72B) | **51.7** | 73.3 | **100.0** | **100.0** |
| | XRayCLIP | Text-Only (Qwen-72B) | Text-Only (MedGemma-27B) | 26.7 | 58.3 | 90.0 | 93.3 |
| | MedSigLIP | Text-Only (MedGemma-27B) | Text-Only (MedGemma-27B) | 30.0 | 53.3 | 83.3 | **100.0** |
| | XRayCLIP | Vision-Language (MedGemma-27B) | Text-Only (MedGemma-27B) | 26.7 | 48.3 | 85.0 | 90.0 |
| | XRayCLIP | Text-Only (Qwen-72B) | Text-Only (Qwen-72B) | 28.3 | 46.7 | 98.3 | 98.3 |
| | OpenCLIP | Text-Only (MedGemma-27B) | Text-Only (MedGemma-27B) | 28.3 | 46.7 | 88.3 | 98.3 |
| | OpenCLIP | Text-Only (MedGemma-27B) | Text-Only (Qwen-72B) | 36.7 | 45.0 | 98.3 | **100.0** |
| | MedSigLIP | Text-Only (MedGemma-27B) | Text-Only (Qwen-72B) | 36.7 | 43.3 | 98.3 | **100.0** |
| | MedSigLIP | Text-Only (Qwen-72B) | Text-Only (MedGemma-27B) | 16.7 | 35.0 | 86.7 | 98.3 |

*Table 5.* We evaluate various image embedding models, alignment scorers, and summarizers on the performance of Stage 2 of SYMBAL.

| | Img Embedding | Alignment Scorer | Summarizer | Reference-Free | | Reference-Based | |
|---|---|---|---|---|---|---|---|
| | | | | Acc@1 | Acc@5 | Acc@1 | Acc@5 |
| Natural | OpenCLIP | Vision-Language (Qwen-72B) | Text-Only (Qwen-72B) | **49.7** | **69.7** | 41.9 | 52.2 |
| | OpenCLIP | Embedding (OpenCLIP) | Vision-Language (Qwen-72B) | 48.1 | 63.9 | 42.5 | 55.6 |
| | OpenCLIP | Embedding (OpenCLIP) | Text-Only (Qwen-72B) | 47.8 | 62.8 | 43.9 | 55.8 |
| | OpenCLIP | Vision-Language (Qwen-72B) | Vision-Language (Qwen-72B) | 45.8 | 62.5 | 38.9 | 52.2 |
| | DINOv2 | Vision-Language (Qwen-72B) | Text-Only (Qwen-72B) | 45.3 | 61.4 | 38.6 | 54.7 |
| | DINOv2 | Text-Only (Qwen-72B) | Text-Only (Qwen-72B) | 43.1 | 60.8 | 41.1 | 56.4 |
| | OpenCLIP | Text-Only (Qwen-72B) | Text-Only (Qwen-72B) | 48.1 | 60.6 | **45.6** | 58.1 |
| | OpenCLIP | Text-Only (Qwen-72B) | Vision-Language (Qwen-72B) | 44.2 | 60.3 | 43.9 | **56.7** |
| | DINOv2 | Text-Only (Qwen-72B) | Vision-Language (Qwen-72B) | 43.6 | 59.7 | 39.7 | 54.2 |
| | DINOv2 | Embedding (OpenCLIP) | Vision-Language (Qwen-72B) | 43.6 | 59.4 | 39.7 | 53.3 |
| Medical | XRayCLIP | Embedding (MedSigLIP) | Vision-Language (MedGemma-27B) | 11.7 | **36.7** | 28.3 | 53.3 |
| | MedSigLIP | Embedding (MedSigLIP) | Vision-Language (MedGemma-27B) | 11.7 | 31.7 | 25.0 | 46.7 |
| | OpenCLIP | Embedding (MedSigLIP) | Vision-Language (MedGemma-27B) | 13.3 | 28.3 | 20.0 | 46.7 |
| | MedSigLIP | Embedding (XRayCLIP) | Vision-Language (MedGemma-27B) | 10.0 | 28.3 | 33.3 | 60.0 |
| | XRayCLIP | Vision-Language (MedGemma-27B) | Vision-Language (MedGemma-27B) | 6.7 | 28.3 | **43.3** | **65.0** |
| | MedSigLIP | Text-Only (MedGemma-27B) | Vision-Language (MedGemma-27B) | 8.3 | 26.7 | **43.3** | **65.0** |
| | OpenCLIP | Text-Only (MedGemma-27B) | Vision-Language (MedGemma-27B) | 10.0 | 25.0 | 23.3 | 63.3 |
| | OpenCLIP | Text-Only (Qwen-72B) | Vision-Language (MedGemma-27B) | 3.3 | 25.0 | 30.0 | 61.7 |
| | MedSigLIP | Embedding (MedSigLIP) | Text-Only (Qwen-72B) | **15.0** | 25.0 | 15.0 | 40.0 |
| | OpenCLIP | Embedding (MedSigLIP) | Text-Only (Qwen-72B) | 13.3 | 23.3 | 16.7 | 48.3 |

**Ablation study.** We now ablate the role of the grouping step across the subset of 360 natural image datasets in our benchmark. We compare SYMBAL to a version that omits grouping: we use the best performing scorer (vision-language scorer with Qwen-72B) in order to flag each individual sentence as valid (1) or misaligned (0), and we then use our best performing summarizer (text-only summarizer with Qwen-72B) in order to identify the unifying concept across the sentences marked as misaligned. All other settings (e.g. prompts, compute budget, model configurations, etc.) are kept identical to those used for SYMBAL. For Stage 1, in the reference-free setting, we observe an Acc@1 of 41.9 and an Acc@5 of 65.3; these metrics represent a substantial decrease from the results obtained with SYMBAL (Acc@1 = 92.8 and Acc@5 = 94.2) in Table 1. We then use the best performing summarizer to identify image features associated with the misaligned sentences. For Stage 2, in the reference-free setting, we observe an Acc@1 of just 3.6 and an Acc@5 of 16.9; again, these are a substantial decrease from the results obtained with SYMBAL (Acc@1 = 49.7 and Acc@5 = 69.7) in Table 2. These results demonstrate the importance of our multi-step, structured approach for addressing the systematic misalignment detection task.

*Table 6.* End-to-end performance across SYMBALBENCH, stratified by domain.

| | Method | Reference-Free | | Reference-Based | |
|---|---|---|---|---|---|
| | | Acc@1 | Acc@5 | Acc@1 | Acc@5 |
| Natural | Llama3.3 70B | 0.3 | 0.3 | 0.6 | 1.4 |
| | Qwen2.5-VL 72B | 0.0 | 1.9 | 0.6 | 1.1 |
| | GPT-OSS 120B | 9.2 | 13.9 | 10.8 | 17.2 |
| | SYMBAL (Ours) | **49.2** | **69.7** | **41.1** | **51.9** |
| Medical | Llama3.3 70B | 0.0 | 8.3 | 0.0 | 5.0 |
| | MedGemma 27B | 0.0 | 1.7 | 0.0 | 0.0 |
| | Qwen2.5-VL 72B | 3.3 | 5.0 | 0.0 | 1.7 |
| | GPT-OSS 120B | 1.7 | 21.7 | 0.0 | 11.7 |
| | SYMBAL (Ours) | **6.7** | **28.3** | **25.0** | **48.3** |

# E. Evaluating SYMBAL in the Wild

In this section, we further demonstrate the utility of SYMBAL by supplementing our evaluations on SYMBALBENCH with additional quantitative and qualitative analyses in real-world settings.

**SYMBAL can accurately surface systematic misalignments in captions generated by off-the-shelf MLLMs.** Below, we list several examples of systematic misalignments identified by SYMBAL, and we also provide associated validation:

- *Example 1:* In captions generated by Llava1.5-7B, SYMBAL detects that erroneous references to a TV ($\hat{t}$) in captions are often systematically associated with the presence of a desk, computer monitor, and/or keyboard ($\hat{v}$) in the scene. We provide visual examples of image-caption pairs with the SYMBAL-identified systematic misalignment in Figure 9 (Row 1). Quantitatively, our analysis finds that erroneous references to a TV in model-generated captions are indeed 13.5 times more likely when a desk is present in the image compared to when a desk is absent, validating the SYMBAL prediction.

- *Example 2:* In captions generated by Llava1.5-7B, SYMBAL detects that erroneous references to a handbag or a handbag on the ground ($\hat{t}$) in captions are often systematically associated with the presence of a bus ($\hat{v}$) in a scene. We provide visual examples of image-caption pairs with the SYMBAL-identified systematic misalignment in Figure 9 (Row 2). Quantitatively, our analysis finds that erroneous references to a handbag in model-generated captions are indeed 3.1 times more likely when a bus is present in the image compared to when a bus is absent, validating the SYMBAL prediction.

- *Example 3:* In captions generated by Llava1.5-7B, SYMBAL detects that erroneous references to a chair ($\hat{t}$) in captions are often systematically associated with the presence of a television ($\hat{v}$) in a scene. We provide visual examples of image-caption pairs with the SYMBAL-identified systematic misalignment in Figure 9 (Row 3). Quantitatively, our analysis finds that erroneous references to a chair in model-generated captions are indeed 3.1 times more likely when a television is present in the image compared to when a television is absent, validating the SYMBAL prediction.

- *Example 4:* In captions generated by Llava1.5-13B, SYMBAL detects that erroneous references to a TV ($\hat{t}$) in captions are often systematically associated with the presence of a computer monitor, keyboard, and/or mouse ($\hat{v}$) in a scene. Interestingly, this systematic misalignment is nearly identical to one that exists in Llava1.5-7B-generated captions (see Example 1), suggesting that solely increasing the scale of the underlying MLLM is insufficient for resolving systematic misalignments. We provide visual examples of image-caption pairs with the SYMBAL-identified systematic misalignment in Figure 10 (Row 1). Quantitatively, our analysis finds that erroneous references to a TV in model-generated captions are indeed 22.2 times more likely when a computer monitor is present in the image compared to when a computer monitor is absent, validating the SYMBAL prediction.

- *Example 5:* In captions generated by LlavaOneVision-7B, SYMBAL detects that erroneous references to text ($\hat{t}$) in captions are often systematically associated with the presence of a sign ($\hat{v}$) in a scene. This systematic misalignment suggests that LlavaOneVision-7B struggles with OCR capabilities, where the presence of text-based signage in an image is likely to result in errors in the generated caption. We provide visual examples of image-caption pairs with the

SYMBAL-identified systematic misalignment in Figure 10 (Row 2). Quantitatively, our analysis finds that erroneous references to text in model-generated captions are indeed 4.6 times more likely when a sign is present in the image compared to when a sign is absent, validating the SYMBAL prediction.

- *Example 6:* In captions generated by AyaVision-8B, SYMBAL detects that erroneous references to a vase $(\hat{t})$ in captions are often systematically associated with the presence of a couch $(\hat{v})$ in a scene. We provide visual examples of image-caption pairs with the SYMBAL-identified systematic misalignment in Figure 10 (Row 3). Quantitatively, our analysis finds that erroneous references to a vase in model-generated captions are indeed 17.7 times more likely when a couch is present in the image compared to when a couch is absent, validating the SYMBAL prediction.

Across all six examples of SYMBAL-identified systematic misalignments provided above, we find that erroneous references to $\hat{t}$ are substantially more likely when $\hat{v}$ is present in the image compared to when $\hat{v}$ is absent. This analysis validates discovered misalignments by demonstrating that links between SYMBAL-identified erroneous textual fact $\hat{t}$ and SYMBAL-identified visual feature $\hat{v}$ do indeed exist.

Our quantitative validation procedure relies on automated annotation methods in order to enable evaluation at scale; in particular, we leverage Qwen-72B in order to annotate erroneous references to $\hat{t}$ in each caption. We find that these generated annotations align closely with human judgments. Given the set of 215 images in the dataset containing a "bus", we tasked a human reader with identifying whether each Llava1.5-7B-generated caption contained an erroneous reference to a "handbag" and/or "handbag on the ground" (Example 2). Human judgments aligned perfectly with Qwen-72B predictions in 96.3% of cases (Cohen's kappa = 0.86).

**SYMBAL is a powerful tool for auditing open-source vision-language datasets.** Below, we list several examples of systematic misalignments identified by SYMBAL on the ShareGPT4V dataset, and we also provide associated validation:

- *Example 7:* SYMBAL detects that erroneous references to a white tablecloth $(\hat{t})$ in captions are often systematically associated with the presence of a table, cake, and/or people $(\hat{v})$ in the scene. We provide visual examples of image-caption pairs with the SYMBAL-identified systematic misalignment in Figure 11 (Row 1). Quantitatively, our analysis finds that erroneous references to a white tablecloth in model-generated captions are indeed 17.2 times more likely when a table is present in the image compared to when a table is absent, validating the SYMBAL prediction.

- *Example 8:* SYMBAL detects that erroneous references to a printer $(\hat{t})$ in captions are often systematically associated with the presence of a computer monitor $(\hat{v})$ in a scene. We provide visual examples of image-caption pairs with the SYMBAL-identified systematic misalignment in Figure 11 (Row 2). Quantitatively, our analysis finds that erroneous references to a printer in model-generated captions are indeed 121 times more likely when a computer monitor is present in the image compared to when a computer monitor is absent, validating the SYMBAL prediction.

- *Example 9:* SYMBAL detects that erroneous references to a black phone $(\hat{t})$ in captions are often systematically associated with the presence of a laptop $(\hat{v})$ in a scene. We provide visual examples of image-caption pairs with the SYMBAL-identified systematic misalignment in Figure 11 (Row 3). Quantitatively, our analysis finds that erroneous references to a black phone in model-generated captions are indeed 48.5 times more likely when a laptop is present in the image compared to when a laptop is absent, validating the SYMBAL prediction.

**Images**

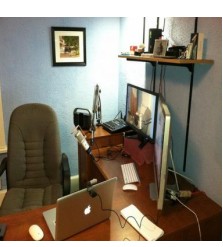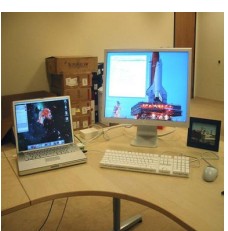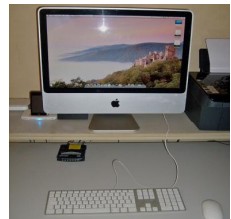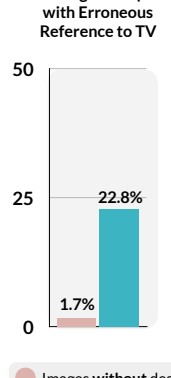

**Llava1.5-7B Generated Captions**

Percentage of Captions with Erroneous Reference to TV

1.7%  22.8%

Images **without** desk
Images **with** desk

The image features a well-organized home office with a desk and a computer setup. On the desk, there is a laptop computer, a keyboard, and a mouse. The desk is accompanied by a chair, which is positioned in front of the desk. In the room, there are several books scattered around, with some on the floor and others on the desk. A potted plant is placed in the corner of the room, adding a touch of greenery to the space. Additionally, A TV is mounted on the wall, providing entertainment options for the home office.

The image displays a small, cluttered office space with a wooden desk. On the desk, there is a laptop computer, a keyboard, and a mouse. A monitor is also present, positioned above the laptop. A chair is placed in front of the desk, ready for someone to sit and work. The office is filled with various items, including a TV mounted on the wall, a cell phone, and a couple of books. There are also two remote controls placed on the desk. In addition to the electronics, there are two bottles and a cup, likely containing beverages for the person working in the office

The image features a wooden desk with a laptop and a desktop computer sitting on it. The laptop is placed on the left side of the desk, while the desktop computer is situated towards the right. Both computers are connected to the same keyboard, which is located in the middle of the desk. Additionally, there is a mouse on the right side of the desk, and a cell phone can be seen on the left side. A TV is mounted on the wall above the desk, providing a source of entertainment or information for the user

The image features a large computer monitor sitting on a desk, with a keyboard and mouse placed in front of it. The monitor is turned on, displaying a beautiful landscape image. The keyboard and mouse are positioned close to the monitor, indicating that they are ready for use. In addition to the main computer setup, there are several other items on the desk. A cell phone can be seen on the left side of the desk, while a book is placed towards the right side. A remote control is also present, located near the center of the desk. A TV is mounted on the wall above the desk, providing a more comprehensive entertainment setup

---

**Images**

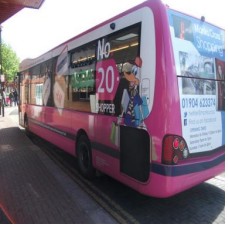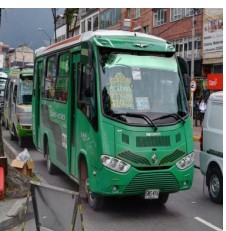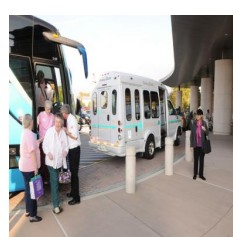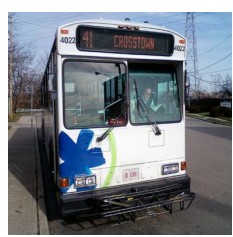

**Llava1.5-7B Generated Captions**

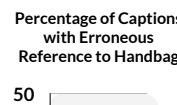

Percentage of Captions with Erroneous Reference to Handbag

5.3%  16.7%

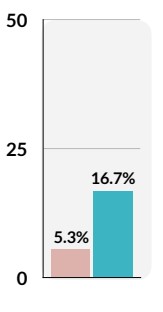

Images **without** bus
Images **with** bus

The image features a pink bus parked on the side of a street, likely at a bus stop. The bus is adorned with a large advertisement on its side, possibly promoting a shopping center. Several people can be seen around the bus, with some standing closer to the bus stop and others further away. There are two handbags visible in the scene, one near the center of the image and the other closer to the right side. The presence of these handbags suggests that the people might be waiting for the bus or have just disembarked from it.

The image features a busy street with a green bus driving down the road, surrounded by various vehicles. There are two other buses in the scene, one on the left side and another on the right side of the street. A truck can also be seen on the right side of the street. Several people are present in the scene, with some standing near the buses and others walking along the street. There are two traffic lights visible in the image, one on the left side and another on the right side of the street, ensuring the smooth flow of traffic. Additionally, there are two handbags in the scene, one near the left side of the street and the other closer to the center.

The image features a group of people standing in front of a bus, likely waiting to board or disembark. There are at least nine people visible in the scene, with some standing closer to the bus and others further away. The bus is parked on the left side of the image, occupying a significant portion of the scene. In addition to the bus, there are several cars visible in the background, with one car on the left side of the image and two cars on the right side. A handbag can be seen placed on the ground near the center of the scene, possibly belonging to one of the passengers.

The image features a white and blue city bus parked on the side of a street. The bus is stopped at a bus stop, and the driver is visible through the windshield. The bus stop has a bench nearby, providing a place for passengers to wait. There are several people in the scene, with one person standing close to the bus, another person further away, and two more individuals closer to the edge of the image. A handbag can be seen placed on the ground near the first person, possibly belonging to a passenger waiting for the bus.

---

**Images**

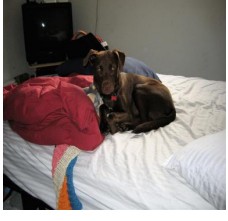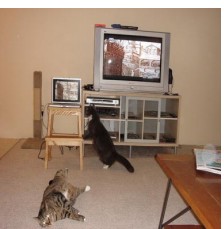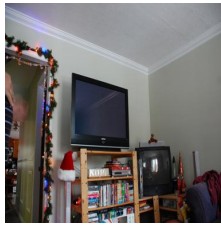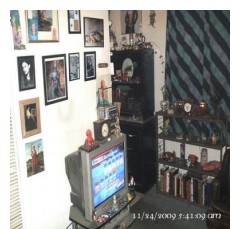

**Llava1.5-7B Generated Captions**

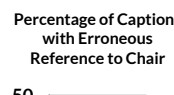

Percentage of Captions with Erroneous Reference to Chair

11.1%  34.3%

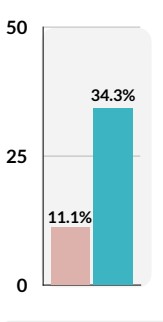

Images **without** TV
Images **with** TV

The image features a brown dog sitting on a messy bed, surrounded by blankets and pillows. The dog appears to be looking at the camera, possibly posing for a picture. The bed is covered with various pillows and blankets, creating a cozy and comfortable environment for the dog. In the room, there is a television on the left side, and a remote control can be seen nearby. A chair is also present in the scene, positioned close to the bed.

The image features a living room with a television set placed on a wooden stand. A cat is lying on the floor, appearing to be watching the TV. Another cat is sitting on the floor, seemingly engaged with the television as well. In the room, there are two chairs, one located near the left side of the television and the other closer to the right side. A remote control can be seen on the floor, possibly used to operate the television. Additionally, there is a book placed on the right side of the room, and a vase is located near the center of the scene.

The image features a living room with a television set placed on a wooden stand. The TV is positioned in the center of the room, surrounded by various books on a bookshelf. The bookshelf is filled with numerous books, creating a cozy atmosphere. In addition to the television and bookshelf, there are two chairs in the room, one located on the left side and the other on the right side. A person can be seen in the room, standing near the left side of the television. The room also has a Christmas tree, adding a festive touch to the space.

The image features a cluttered living room with a television set placed on a stand in the center. The room is filled with various items, including a large collection of books scattered throughout the space. Some books are placed on the floor, while others are stacked on shelves or placed on surfaces. In addition to the books, there are several figurines and knick-knacks, such as a clock, a vase, and a cup, adding to the cluttered appearance of the room. A chair can be seen in the background, and a potted plant is placed near the right side of the room. The overall atmosphere of the living room is busy and filled with various items, creating a cozy yet disorganized space

*Figure 9.* Examples of image-caption pairs with SYMBAL-identified systematic misalignments are shown here, with the identified erroneous textual fact in each caption highlighted in red. We also quantitatively validate each identified systematic misalignment. [Row 1] SYMBAL detects that erroneous references to a TV ($\hat{t}$) in captions are often systematically associated with the presence of a desk, computer monitor, and/or keyboard ($\hat{v}$) in the scene. [Row 2] SYMBAL detects that erroneous references to a handbag or handbag on the ground ($\hat{t}$) in captions are often systematically associated with the presence of a bus ($\hat{v}$) in a scene. [Row 3] SYMBAL detects that erroneous references to a chair ($\hat{t}$) in captions are often systematically associated with the presence of a television ($\hat{v}$) in a scene.

Images

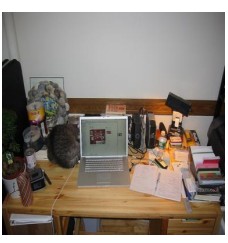 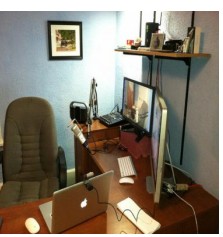 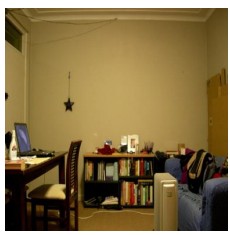 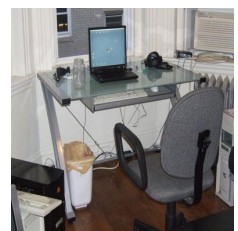

**Percentage of Captions with Erroneous Reference to TV**

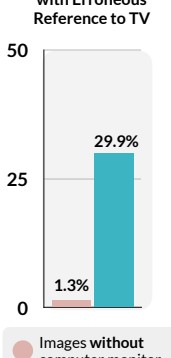

29.9%

1.3%

- Images **without** computer monitor
- Images **with** computer monitor

Llava1.5-13B Generated Captions

The image features a wooden desk with a laptop computer sitting on top of it. A cat is comfortably resting on the desk, close to the laptop. The desk is surrounded by various items, including several books scattered around the area. In addition to the books, there are two bottles on the desk, one near the left side and the other near the right side. A potted plant is located on the left side of the desk, adding a touch of greenery to the scene. **A TV can be seen in the background, mounted on the wall above the desk.**

The image depicts a home office setup with a desk featuring a laptop and a desktop computer. The laptop is placed on the left side of the desk, while the desktop computer is positioned on the right side. A keyboard is situated in front of the laptop, and a mouse can be found on the left side of the desk. A chair is situated in front of the desk, ready for someone to sit and work. **A TV is mounted on the wall above the desk, providing additional entertainment or information during work hours.** In the room, there are also a few books scattered around, possibly for reference or leisure reading. A person can be seen in the background, possibly working or observing the office setup.

The image depicts a small, cluttered living room with various pieces of furniture and items. There is a couch situated in the background, and a chair placed in the middle of the room. A dining table is located near the left side of the room, with a laptop and a mouse placed on it. A bookshelf filled with numerous books is also present in the room, adding to the cluttered appearance. In addition to the books, there are several other items scattered throughout the room, such as a backpack, a handbag, a bottle, and a cup. **A TV can be seen mounted on the wall**, and a clock is hanging nearby. The room also features a window, providing natural light to the space.

The image features a small office space with a desk and a computer setup. On the desk, there is a laptop computer, a keyboard, and a mouse. A chair is positioned in front of the desk, ready for someone to sit and work. In addition to the computer setup, there are two bottles placed on the desk, one near the laptop and the other closer to the edge of the desk. **A TV is mounted on the wall above the desk**, and a book can be seen on the desk as well. A remote control is also present in the scene, likely for the TV.

---

Images

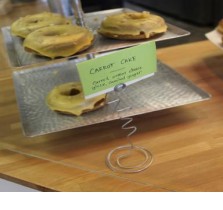 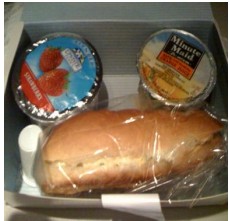 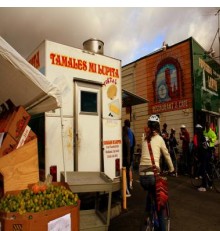 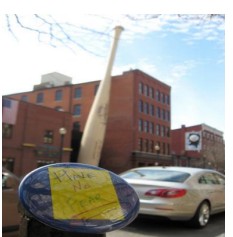

**Percentage of Captions with Erroneous Reference to Text**

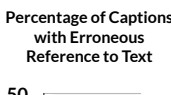
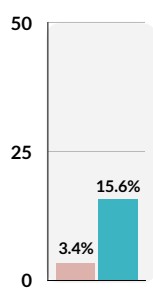

15.6%

3.4%

- Images **without** signs
- Images **with** signs

LlavaOneVision-7B Generated Captions

The image depicts a display of doughnuts on a metal tray, placed on a wooden surface. The doughnuts are frosted with a yellow glaze and appear to be of a similar size and shape. **In the foreground, there is a green sign with white text that reads "Carrot Cake" and lists additional ingredients: "Carrot, cinnamon, nutmeg, ginger, and cream cheese."** The sign is attached to a metal stand with a coiled wire design. The background is slightly blurred but shows what appears to be a kitchen or bakery setting with stainless steel surfaces and a person partially visible. The lighting in the image is bright, highlighting the doughnuts and sign.

The image depicts a meal consisting of a sandwich, two containers of yogurt, and a small packet of salt. The sandwich is placed in the center of the image, wrapped in plastic wrap, and appears to be a sub-style sandwich with a light brown crust. To the left of the sandwich, there is a container of yogurt with a blue lid, featuring an image of strawberries and the text "STRAWBERRY" in white and red letters. **To the right of the sandwich, there is another container of yogurt with a yellow lid, labeled "Minute Maid" and "SWEET TARTS" in black and red letters.** The yogurt container also has an image of a grape. In the bottom left corner of the image, there is a small white packet of salt...

The image depicts a lively street scene with a food truck prominently featured. The truck is white with red and yellow lettering that reads "TAMALELES MI LUPITA. **" Below the main text, there is a smaller sign that says "TAMALEES," and additional text that appears to be a menu or description of the food offered.** The truck has a window for service, and there is a logo or emblem above the window, which includes a depiction of a tamale and a corn husk. In front of the truck, there is a wooden crate filled with green limes, and a person is seen riding a bicycle past the truck. The cyclist is wearing a helmet and a backpack, and the bicycle has a basket on the front...

The image depicts a tall, white baseball bat leaning against a brick building. In the foreground, there is a round object with a yellow sticky note attached to it. **The note has handwritten text that reads "HATE NO FEAR."** The sky is partly cloudy, and there are trees with bare branches visible in the upper left corner of the image. In the background, there are parked cars and a street lamp. The overall scene appears to be set in an urban environment, possibly a street or a public area near a sports facility or a baseball-themed attraction.

---

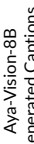Images

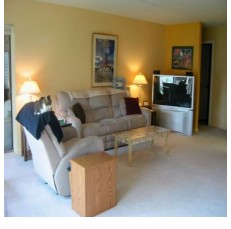 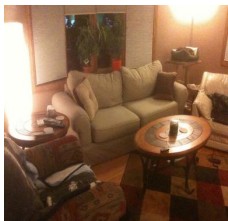 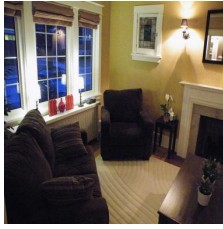 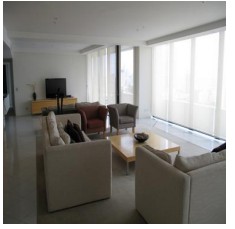

**Percentage of Captions with Erroneous Reference to Vase**

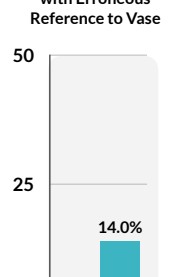

14.0%

0.8%

- Images **without** couch
- Images **with** couch

Aya-Vision-8B Generated Captions

The image depicts a cozy living room with warm, inviting decor. The walls are painted a soft yellow, complementing the beige carpet that covers the floor. A plush, light-colored sofa is positioned against one wall, adorned with a few throw pillows, including a red accent pillow. A recliner sits nearby, offering a comfortable spot to relax, with a black cat perched on its armrest. **In front of the sofa, a small wooden coffee table holds a decorative vase and a few items.** Across from the sofa, a television is mounted on the wall above a wooden entertainment console. Two floor lamps with warm-toned shades flank the television, providing additional lighting...

This cozy living room is bathed in warm, ambient lighting, creating a welcoming atmosphere. A plush, light-colored sofa sits centrally, adorned with soft throw pillows and cushions, inviting relaxation. **Across from the sofa, a wooden coffee table with a circular glass top holds a few items, including a salt shaker and a small vase with dried flowers.** To the left, a side table supports a lamp with a warm-toned shade, casting a gentle glow. A potted plant adds a touch of greenery to the room, while a rug with a bold geometric pattern covers part of the hardwood floor. In the background, a window with sheer curtains allows natural light to filter in...

This cozy living room features warm, inviting decor with a touch of elegance. The space is illuminated by soft lighting, creating a relaxing atmosphere. A dark brown sofa with striped cushions sits against the far wall, accompanied by a matching armchair positioned near a small side table. **A coffee table sits in the center, adorned with a vase of red roses.** The room is decorated with a mix of traditional and modern elements, including a fireplace with a mantelpiece and a wall-mounted lamp. Large windows frame the room, allowing natural light to filter in during the day and soft glow from outside lights at night...

This image showcases a modern and minimalist living room designed with a clean, white aesthetic. The space features a large window with sheer white curtains, allowing natural light to flood the room. In the center, there is a light-colored sectional sofa accompanied by a cream-colored loveseat and a red armchair, creating a cozy yet stylish seating area. **A low, rectangular coffee table sits in front of the sofa, adorned with a vase of flowers.** Across from the seating area, a flat-screen television is mounted on the wall, positioned to face the seating arrangement. The room has a sleek, open layout with a hardwood floor and white walls, enhancing the sense of spaciousness...

*Figure 10.* Examples of image-caption pairs with SYMBAL-identified systematic misalignments are shown here, with the identified erroneous textual fact in each caption highlighted in red. We also quantitatively validate each identified systematic misalignment. [Row 1] SYMBAL detects that erroneous references to a TV ($\hat{t}$) in Llava1.5-13B-generated captions are often systematically associated with the presence of a computer monitor, keyboard, and/or mouse ($\hat{v}$) in the scene. [Row 2] SYMBAL detects that erroneous references to text ($\hat{t}$) in LlavaOneVision-7B-generated captions are often systematically associated with the presence of a sign ($\hat{v}$) in a scene. [Row 3] SYMBAL detects that erroneous references to a vase ($\hat{t}$) in AyaVision-8B-generated captions are often systematically associated with the presence of a couch ($\hat{v}$) in a scene.

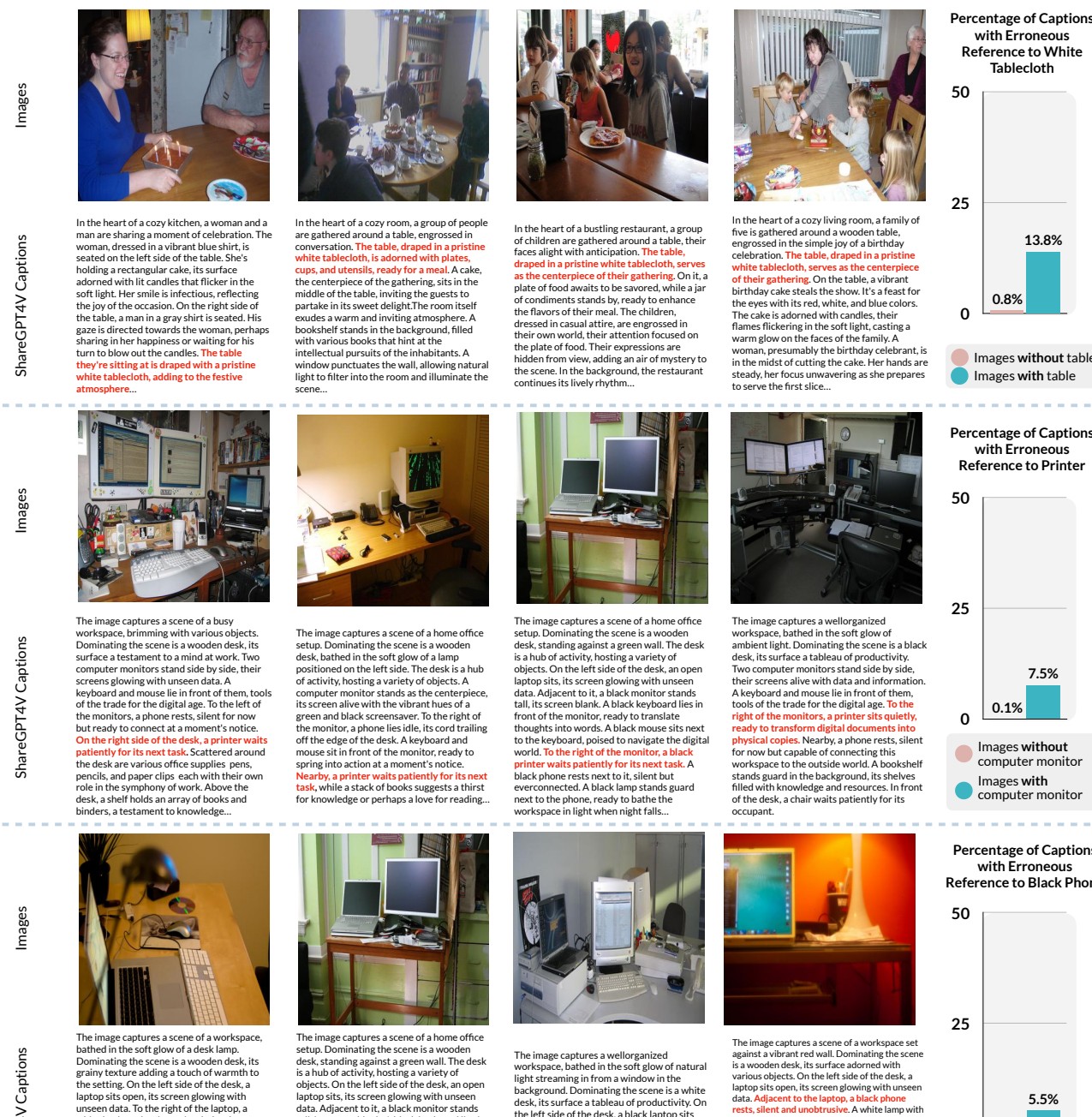

*Figure 11.* Examples of image-caption pairs with SYMBAL-identified systematic misalignments are shown here, with the identified erroneous textual fact in each caption highlighted in red. We also quantitatively validate each identified systematic misalignment. [Row 1] SYMBAL detects that erroneous references to a `white tablecloth` ($\hat{t}$) in ShareGPT4V captions are often systematically associated with the presence of a `table`, `cake`, and/or `people` ($\hat{v}$) in the scene. [Row 2] SYMBAL detects that erroneous references to a `printer` ($\hat{t}$) in ShareGPT4V captions are often systematically associated with the presence of a `computer monitor` ($\hat{v}$) in a scene. [Row 3] SYMBAL detects that erroneous references to a `black phone` ($\hat{t}$) in ShareGPT4V captions are often systematically associated with the presence of a `laptop` ($\hat{v}$) in a scene.

