# OpenReview forum: "Symbal: Detecting Systematic Misalignments in Model-Generated Captions"
_ICML.cc/2026/Conference — ICML 2026 regular_

### Official Review · Reviewer_PLXF · 2026-03-10

**Soundness:** 2
**Presentation:** 3
**Significance:** 2
**Originality:** 3
**Overall Recommendation:** 4
**Confidence:** 4

**Summary:**

This paper introduces the task of systematic misalignment detection for MLLM-generated captions, where an erroneous textual fact is repeatedlly associated with the specific viusal features. The author proposes Symbal, which structures the task into two sequential stages: Symbal first detects recurring textual errors in captions and then identifies the associated visual feature in the second stage. The paper also introduces SymbalBench, a synthetic benchmark of 420 evaluation settings with planted misalignments across natural (COCO) and medical (MIMIC-CXR) domains.

**Compliance With Llm Reviewing Policy:**

Affirmed.

**Final Justification:**

The author's reply basically resolved my concerns.

**Key Questions For Authors:**

1.Systematic misalignments can be plausible and hard to detect. If errors driven by real co-occurrence between visual features and concepts,  the scorer—particularly if it is also an MLLM-as-a-judge—may share similar biases with the captioning model, leading to systematic false negatives. The paper relies mainly on score averaging and a stronger scorer, but lacks direct  robustness analysis.

2.Given that medical-domain performance is clearly weaker than in natural images and reference-free settings are especially challenging. Can the authors report application-oriented metrics for real auditing utility, such as human-in-the-loop precision/recall for discovered $(t,v)$ , false-alarm cost, and whether the findings help data cleaning or model improvement?

3.The method represents visual features as "single-phrase, directly visible" concepts. This is interpretable and easy to evaluate, but medical cues are often subtle (findings, location, severity, or morphology) and may not fit a short phrase. This likely contributes to the weaker medical-domain results compared to natural images.

**Limitations:**

yes

**Strengths And Weaknesses:**

Strength

1.The task formulation and benchmark construction are novel and well-motivated. The paper formalizes "systematic misalignment" as cross-modal pairs $(t,v)$  and provides a scalable synthetic benchmark for evaluation.

2.The paper decomposes "systematic misalignment detection" into a clear two-stage pipeline. The modular grouping–scoring–summarizing design is practical and easy to implement.

3.The experiments include relatively thorough component ablations, showing how different scorers / summarizers / embeddings affect performance.

Weakness

1.The reported 63.8% Acc@5 is obtained on the synthetic SYMBALBENCH with author-injected errors, so it may not reflect performance on naturally occurring, more subtle real-world misalignments. More importantly, the study used artificially synthesized error data instead of errors collected from existing models, which reduces the reliability of the study's conclusions.

2.The main baseline is "directly prompting an LLM to output $(t,v)$  from the dataset," but due to context window limits it relies on sampling. The paper should more clearly specify the sampling size and strategy, whether random seeds are fixed, and how "compute fairness" is measured when comparing to SYMBAL.

3.The method treats sentence-level splits of captions as the basic "textual fact" units. This design choice strongly affects which error types can be detected, yet the paper provides limited justification, few alternatives, and little discussion of failure modes or boundary conditions.

---

> ### Author Rebuttal · Authors · 2026-03-31
>
> We thank Reviewer PLXF for reviewing our work and providing helpful feedback.
>
> > [Q1] Synthetic Data.
>
> We refer the reviewer to our response to Reviewer QUK2 [Q1], where we address this point.
>
> > [Q2] Baseline.
>
> For our direct prompting baselines, we considered three LLMs with comparable parameter counts to Symbal. To ensure compute fairness, all baseline experiments were conducted on the same hardware as Symbal (i.e. max 2 NVIDIA H100 80GB GPUs). We used random sampling without replacement to select 200 samples from each of the 420 datasets in SymbalBench, the maximum value that would fit in both (1) the context window of the LLMs as well as (2) the available compute. We intentionally did not fix the random seed across the 420 datasets in SymbalBench; this ensures that our reported results reflect the aggregate expected performance across the entire benchmark distribution, rather than being skewed by a specific 200-sample slice. Prompts are in Appendix E.  We will update the manuscript to include these details.
>
> > [Q3] Textual Units.
>
> We opted to split long-form captions at the sentence-level, as a sentence typically refers to a self-contained fact. This strategy is used commonly in prior works in both the medical and natural domains (e.g. Zhang et al., ML4H 2022), and splitting at this level allows Symbal to isolate precise errors. During development, we also considered (1) word/phrase-level units, which often lacked sufficient semantic context for our scorers and summarizers, (2) full-caption units, where signal was often diluted, and (3) using an LLM to restructure captions into atomic facts, which often introduced additional errors. Overall, we find that sentence-level splits provide an optimal signal-to-noise ratio for systematic error detection.
>
> We acknowledge that this design choice has boundary conditions that may affect performance, such as coreference resolution issues; we will update our manuscript to discuss these trade-offs. We also note that users of Symbal can easily adjust this design choice depending on their use-case.
>
> > [Q4] Similar Biases.
>
> While it is certainly possible for Symbal's constituent models to capture similar biases as the original MLLM used for caption generation, we emphasize that the target tasks are different: the original MLLM is used for generation, whereas Symbal is used for verification. The verification procedure employed by Symbal, which involves evaluating alignment between image-text pairs and summarizing results, utilizes a fundamentally different reasoning process when compared to open-ended caption generation; as a result, this procedure will be more robust to shared biases. This claim is supported by several recent works (e.g. Lee et al., NAACL 2024; Guan et al. NAACL 2024); these studies suggest that even when identical model backbones are used for generation and verification, shared biases have a limited impact on verification abilities.
>
> > [Q5] Medical Domain Performance.
>
> We thank the reviewer for this suggestion and agree that application-oriented metrics for auditing utility are important. While a full reader study is beyond our current scope, our paper provides substantial evidence of practical utility:
> - False Alarm Cost: The high Acc@5 metrics in the natural domain show that a human auditor would consistently find a true systematic misalignment within Symbal's top predictions, significantly reducing the search cost when compared to manual auditing. In the medical domain, the Acc@5 metrics are indeed lower due to the complex setting yet Symbal still provides a high-probability starting point (Acc@5=48.3 with references); in contrast, manual auditing introduces "needle-in-a-haystack" challenges as well as requires domain experts.
> - Interpretability: Since we explicitly designed Symbal to output the textual error and visual feature in natural language, the outputs are immediately interpretable and actionable for data cleaning or model improvement.
> - Real-world utility: Our extensive quantitative and qualitative evaluations in Section 5.4 and Appendix F show that Symbal-identified misalignments are accurate in real-world settings and that Symbal enables precise model/dataset auditing.
>
> > [Q6] Medical Concepts.
>
> We agree that visual features in medical images are often complex and subtle in nature. We opted to use a single, short phrase to describe identified visual features in order to ensure that results were interpretable and immediately actionable for users, especially when using Symbal's outputs to clean data or improve models. Importantly, the lower medical-domain performance observed on SymbalBench is not due to this design choice (the benchmark’s ground-truth concepts can be captured with short phrases) but rather reflects the difficulty of the domain. We also emphasize that Symbal provides flexibility: users can easily adjust summarizer prompts to include more granular details, such as anatomical locations or severity, if desired.

---

> > ### Author Rebuttal · Reviewer_PLXF · 2026-04-04
> >
> > Thank you for the response.
> >
> > I remain skeptical about whether using artificially assumed error data—such as assuming an airplane would be misidentified as a "person"—can effectively simulate the actual errors MLLMs encounter in real-world applications.
> >
> > I believe a more reasonable pipeline would involve collecting and extracting errors from actually deployed MLLMs, rather than manually setting up text pairs (as shown in Appendix Table 3) to forcibly map one vocabulary term to another. This approach risks a disconnect where, despite the research motivation being sound, the foundation of the study is built upon a simulated scenario that may fail to reflect the genuine issues present in real MLLMs.
> >
> > The other parts of the response have addressed my concerns.

---

> > > ### Author Response · Authors · 2026-04-05
> > >
> > > We thank Reviewer PLXF for their response and for reviewing our manuscript; however, we believe there may be a misunderstanding regarding the mechanism of error injection in our benchmark, which we would like to clarify.
> > >
> > > First, we emphasize that our benchmark does **not assume that an airplane would be misidentified as a person nor does it forcibly map one vocabulary term to another**; we agree with the reviewer that such a setting would be unrealistic, and we do not use this approach. Rather, our benchmark is designed to simulate systematic misalignments, where we assume that an MLLM consistently inserts a textual error (e.g. the presence of a "person")  when a particular visual trigger is present (e.g. the presence of an "airplane"). For example, as shown in Figure 1, an example caption for an image of an airplane would be, "The photo includes one person, clearly visible. A propellor airplane is on a grassy runway"; here, the simulated systematic misalignment results in the insertion of a plausible textual error ("person") into a pre-existing, accurate caption. While synthetic, our error injections are designed to approximate systematic co-occurrence patterns (e.g. consistent error insertion conditioned on visual triggers), which capture the core structure through which spurious correlations manifest in real MLLM generations. The simulated systematic misalignments in Appendix Table 3 that comprise SymbalBench are based on evidence from prior work (Li et al., 2023; Oakden-Rayner et al., 2020).
> > >
> > > Second, as we discuss in our response to [Q1] above, **a synthetic benchmark is the only scientifically-sound way that detection methods can be quantitatively evaluated**. Ground-truth errors associated with actually-deployed MLLMs are simply unknown. Collecting accurate, comprehensive human labels for this task is impossible; a human reader would need to identify an unknown number of subtle, recurring patterns across hundreds of thousands of dense image-caption pairs. This task exceeds the cognitive capabilities of human readers. Thus, while the reviewer's proposed approach of collecting and extracting errors from actually deployed MLLMs is feasible for sample-level misalignment detection (as we discuss in Appendix A), this approach is not viable when investigating dataset-level, systematic patterns as we do in this work.
> > >
> > > Third, we wish to address the final point from the reviewer by emphasizing that **the simulated scenario does indeed reflect the genuine issues present in real MLLMs**. Crucially, we provide extensive analysis in Section 5.4 and Appendix F that Symbal, which performs well on the simulated benchmark, can also translate to realistic scenarios by identifying multiple, verifiable systematic misalignments across 4 actually-deployed MLLMs. SymbalBench also extends an established line of research; many recent benchmarks have used automated methods to generate datasets with specific pre-defined characteristics, such as injected errors. Some popular examples include the FOIL dataset (Shekhar et al., ACL 2017), the VisDiff dataset (Dunlap et al., CVPR 2024), and the Domino evaluation framework (Eyuboglu et al., ICLR 2022).
> > >
> > > We hope that the responses above answer your questions, and we're happy to provide additional clarifications if necessary. Thank you again for reviewing our paper, and we hope that you will consider increasing your score.

---

### Official Review · Reviewer_1PXJ · 2026-03-13

**Soundness:** 3
**Presentation:** 3
**Significance:** 3
**Originality:** 2
**Overall Recommendation:** 4
**Confidence:** 4

**Summary:**

This paper proposes a framework to detect spurious correlation in generated captions at the dataset level, which they term "systematic misalignment". The framework is composed of two stages: (1) text-level misalignment detection and (2) visual-level misalignment detection. In particular, they first cluster the embeddings (text or vision) and get the top misaligned cluster examined by an alignment scorer such as CLIP. Then, they provide a summarization or explanation of what the cluster means conceptually. Doing this separately for both text and vision modalities, they output the pair of visual and textual explanation (feature or concepts) as the pair of systematic misalignment. The main contribution includes a curated benchmark, with ground-truth and controlled pair of systematic misalignment and the proposed framework.

**Compliance With Llm Reviewing Policy:**

Affirmed.

**Key Questions For Authors:**

See weakness

**Limitations:**

yes

**Strengths And Weaknesses:**

## Strength
* The dataset-level spurious correlation detection is an important problem as more and more synthetically generated captions are being used to train LVLMs. The existence of those problems can propagate to the trained model.
* The curation of the new benchmark does align with the proposed problem setting.
* The result shows that existing real dataset like ShareGPT4V also contains dataset-level spurious correlation.

## Weakness
* The use of the term "systematic misalignment" seems not aligned with the literature, where the definition refers to hallucination caused by spurious correlations, where any of the visual input feature $v_j$ is the spurious attribute, and the textual fact output $t_i$ is the hallucinated output correlated with $v_j$.
* The baseline comparison is not sufficient. The authors claim that this is a new problem setting. However, there already exists a line of work on caption hallucination detection at the sample level. One could definitely aggregate the detection results at the sample level, whether it's a sentence or concept provided by the detector, to produce a meaningfully summarized prediction of systematic misalignment at the dataset level. Or even more simply, one could direct prompt a VLM to detect the hallucination sentence at the sample level, producing a subset of samples having hallucinated sentences. Their associated images can then be summarized as well as the "predicted" hallucinated sentences. This is already a simple baseline that the authors should consider having.

---

> ### Author Rebuttal · Authors · 2026-03-31
>
> We thank Reviewer 1PXJ for reviewing our work and providing helpful feedback.
>
> > [Q1] Terminology.
>
> We thank the reviewer for this observation. We note that prior work on hallucination has primarily focused on per-sample errors rather than consistent, dataset-level patterns. We introduce the term systematic misalignment to refer to systematic, dataset-level dependencies between erroneous textual facts $t$ and visual features $v$. While systematic misalignments do often arise due to spurious correlations, our formulation is broader and meant to capture any recurring error pattern. Our terminology includes cases where the error may not be a traditional hallucination (i.e. an object that isn't there); for instance, in Figure 10 Row 2, Symbal identifies a systematic misalignment associated with the LlavaOneVision-7B model, where erroneous transcriptions of text ($t$) are associated the presence of signage in the scene ($v$).
>
> We refer the reviewer to Appendix A (Related Work) for additional details, where we situate our problem setting in the context of existing work. We will be sure to add these clarifications in the next revision of our paper.
>
> > [Q2] Baselines.
>
> Thank you for this suggestion. We agree that many prior works have investigated hallucination detection at the sample level; we refer the reviewer to Appendix A (Related Work) where we discuss these approaches in detail and highlight our novel contributions.
>
> We note that aggregating outputs of sample-level hallucination detectors in order to identify dataset-level trends is non-trivial. In effect, a proper implementation of this approach would require scoring image-caption pairs and summarizing both text-level trends as well as image-level trends, which effectively reintroduces the core components of Symbal. Thus, the reviewer's proposed baseline can essentially be viewed as a simplified ablation of our method (i.e. Symbal without the clustering step), which we find to be less effective in practice, as shown below.
>
> In order to demonstrate this empirically, we consider the subset of 360 natural image datasets in our benchmark, and we use the best performing scorer (vision-language scorer with Qwen-72B) in order to flag each individual sentence as valid (1) or misaligned (0). We then use our best performing summarizer (text-only summarizer with Qwen-72B) in order to identify the unifying concept across the sentences marked as misaligned. All other settings (e.g. prompts, compute budget, model configurations, etc.) are kept identical to those used in the paper for Symbal. For Stage 1, in the reference-free setting, we observe an Acc@1 of 41.9 and an Acc@5 of 65.3; these metrics represent a substantial decrease from the results obtained with Symbal (Acc@1 = 92.8 and Acc@5 = 94.2) in Table 1. We then use the best performing summarizer to identify image features associated with the misaligned sentences. For Stage 2, in the reference-free setting, we observe an Acc@1 of just 3.6 and an Acc@5 of 16.9; again, these are a substantial decrease from the results obtained with Symbal (Acc@1 = 49.7 and Acc@5 = 69.7) in Table 2. Ultimately, these results show that a multi-step, structured approach for this task is critical.

---

> > ### Author Rebuttal · Reviewer_1PXJ · 2026-04-03
> >
> > The baseline inclusion resolves my concern.

---

> > > ### Author Response · Authors · 2026-04-08
> > >
> > > Thank you for the thoughtful feedback - we're glad that the new baseline has addressed your concerns. We will be sure to include this baseline in our next revision.
> > >
> > > We appreciate the time you've taken to review our paper, and we would be grateful if you would consider increasing your score.

---

### Official Review · Reviewer_QUk2 · 2026-03-13

**Soundness:** 3
**Presentation:** 3
**Significance:** 2
**Originality:** 2
**Overall Recommendation:** 4
**Confidence:** 2

**Summary:**

This work introduces the task of systematic misalignment detection, which aims to identify recurring captioning errors in MLLM-generated image-text datasets that are consistently associated with specific visual features. To address this problem, the authors propose SYMBAL, a dual-stage framework that first discovers recurring textual errors in captions and then identifies the visual features linked to those errors, producing natural-language summaries of the resulting failure modes. They also introduce SYMBAL-BENCH, a benchmark of 420 annotated vision-language datasets spanning natural and medical images, designed to evaluate automated methods on this task. Experiments show that SYMBAL outperforms prior baselines on the benchmark.

**Compliance With Llm Reviewing Policy:**

Affirmed.

**Key Questions For Authors:**

Can the author extends the experiments from #404 10k subset to a much larger set, like 500k?

**Limitations:**

plz check my comments above.

**Strengths And Weaknesses:**

I think the proposed problem is interesting and all the used components sound. Their experiments suggest that the method could help surface plausible but spurious image-caption associations in both model-generated captions and existing vision-language datasets.

I am unsure if the proposed benchmark is well-stood to verify the proposed problem due to its scale and simplicity. For the simplicity, it is the dataset naturally assume there is semantically-similar facts that could yield the described "textual errors". However, for real cases, we have million or even billions image-caption pairs. How could we find such systematical errors inside this kind of dataset -- we have no prior partitions (small images clusters) regarding the dataset. The proposed benchmarks seems couldn't support examing such capabilities.

Also, does the proposed method could scale up while maintain effectiveness? Providing some empirical evidence here could be very helpful.

---

> ### Author Rebuttal · Authors · 2026-03-31
>
> We thank Reviewer QUK2 for reviewing our work and providing helpful feedback.
>
> > [Q1] Benchmark Composition + Simplicity.
>
> Below, we respond to both [Q1] from Reviewer QUK2 as well as [Q1] from Reviewer PLXF. In our response below, we (1) quantitatively demonstrate that our approach extends beyond SymbalBench to real-world settings, (2) clarify the role for SymbalBench as a controlled evaluation testbed, and (3) discuss the complexity of SymbalBench.
>
> **Quantitative validation on real-world data:** We refer the reviewers to our evaluations in Section 5.4 and Appendix F, where we directly show that Symbal identifies complex systematic misalignments and quantitatively validate findings.
>
> Results show that Symbal accurately surfaces non-trivial, previously-unknown systematic errors across multiple real-world models and datasets, demonstrating external validity. As an example, in captions generated by LlavaOneVision-7B, Symbal detects that erroneous descriptions of text in captions are often systematically associated with the presence of a "sign” in a scene (Figure 10, Row 2). Quantitative validation shows that erroneous references to text in  captions are indeed 4.6 times more likely when a sign is present in the image compared to when a sign is absent, validating the Symbal prediction. Similar findings across 4 models and 2 datasets in Appendix F show the reliability of our study’s conclusions and the utility of our approach in the real world.
>
> **Motivation behind SymbalBench:** The key challenge behind evaluating methods like Symbal on real-world vision-language datasets is that ground-truth systematic misalignments are unknown. Collecting human labels here is infeasible, as human cognitive limits make it nearly impossible to identify subtle, recurring patterns across thousands of information-dense image-caption pairs.
>
> Thus, without access to ground-truth annotations, it becomes mathematically impossible to compute metrics like accuracy on real-world data. Thus, SymbalBench is a scientific necessity - by injecting pre-defined systematic misalignments into base vision-language datasets, we can yield a controlled test bed where ground-truth annotations are available. **The automated nature of our approach provides several key advantages, including (1) the ability to operate at scale by generating hundreds of datasets, (2) the presence of ground-truth labels that are guaranteed to be accurate, and (3) the ability to quantitatively assess detection capabilities - a measurement that is essential when comparing algorithms.**
>
> This follows many recent benchmarks generated via automated methods (e.g. FOIL, VisDiff, and Domino).
>
> **Complexity and realism of injected misalignments:** Our automated procedure for injecting pre-defined systematic misalignments is explicitly designed to represent the types of subtle, complex misalignments that are likely to emerge in the wild. Specifically, our benchmark covers (1) multiple domains; (2) diverse misalignments that are plausible in real-world settings as observed by prior works (Li et al., EMNLP 2023, Oakden-Rayner et al. CHIL 2020); (3) a range of misalignment strengths; and (4) a range of visual feature sizes.  We emphasize that the benchmark is indeed challenging, with standard baselines exhibiting poor performance; this demonstrates that this task and benchmark require sophisticated reasoning capabilities in order to be solved effectively, rather than just simple semantic matching.
>
> > [Q2] Benchmark Scale
>
> Symbal can operate effectively on large datasets. Importantly, Symbal does not rely on prior predefined partitions; rather, our grouping approach automatically discovers these partitions by clustering text and visual features in the embedding space. Symbal's components can operate effectively across datasets of arbitrary size and are computationally efficient (i.e. operations can be parallelized and have been optimized using the Faiss and vLLM packages).
>
> In response to the reviewer's suggestion, we have extended our experiments to 100,000 image-caption pairs from the ShareGPT4V dataset. We find that at this larger scale, Symbal (reference-free) continues to identify multiple non-trivial systematic misalignments. As an example, one misalignment identified by Symbal is that erroneous references to a "coffee maker" and "blender" ($\hat{t}$) are often associated with the presence of a "sink", "countertop", and "stove" ($\hat{v}$) in a scene. As a second example, Symbal identifies that erroneous references to a "black stapler" ($\hat{t}$) are often associated with the presence of a "cluttered desk" ($\hat{v}$).
>
> While we run evaluations on 100K samples for this rebuttal, our approach can easily extend to 500K or millions of samples given additional compute. We will update our paper to discuss these additions.

---

> > ### Author Rebuttal · Reviewer_QUk2 · 2026-04-06
> >
> > I appreciate the authors’ efforts in the rebuttal, including their clarification of the contribution and the 100k experiment. However, I do not think these points adequately address my concerns, which would require substantial changes to resolve.
> >
> > I am still unsure if the proposed benchmark is well-stood to verify the proposed problem due to its scale and simplicity, also, could this method effectively scale-up in a practical way.

---

> > > ### Author Response · Authors · 2026-04-07
> > >
> > > We thank reviewer QUK2 for their response and for reviewing our manuscript. We would like to offer a final clarification regarding benchmark scale/complexity and practical scalability.
> > >
> > > **Benchmark Scale:** SymbalBench is not a single, small dataset but rather a large-scale benchmark comprised of 420 datasets. Each of these datasets includes 2K to 5K image-text pairs, which is comparable to the scale of standard validation datasets like COCO and MIMIC-CXR often used for evaluating MLLMs. In total, across all 420 datasets, the **benchmark includes 1.7 million image-text pairs**.
> > >
> > > **Benchmark Complexity:** While SymbalBench is synthetic in nature, it is not simplistic. Our error injections are designed to approximate systematic co-occurrence patterns (e.g. consistent error insertion conditioned on visual triggers), which capture the core structure through which spurious correlations manifest in real MLLM generations.
> > >
> > > We emphasize that controlled benchmarks of this form are the only scientifically-sound way to enable quantitative evaluation on this task. In real-world datasets at the million- or billion-scale, the ground-truth systematic misalignments are unknown, making it impossible to compute metrics like accuracy. SymbalBench addresses this challenge, enabling rigorous and reproducible evaluations.
> > >
> > > Our benchmark covers highly-challenging settings:
> > > - **Multiple domains:** SymbalBench encompasses both natural image and medical image data, representing two distinct domains. Medical imaging data represents a complex and specialized setting, where detection of systematic misalignments is particularly critical prior to real-world deployment.
> > > - **Diverse misalignments that are plausible in real-world settings:** Selected systematic misalignments are grounded in literature (Oakden-Rayner et al., 2020; Li et al., 2023)
> > > - **Varying strengths of the injected systematic misalignment:** In real-world settings, systematic misalignments exhibit variations in strength. In order to reflect this, we consider three possible levels of association between the erroneous textual fact $t$ and visual feature $v$ in our datasets: low association (Cramer’s V = 0.3), moderate association (Cramer’s V = 0.6), and high association (Cramer’s V = 0.9).
> > > - **Visual features of varying sizes:** Visual features in real-world datasets exhibit varying sizes, and in particular, small features present a particularly challenging setting. SymbalBench exhibits substantial coverage across varying sizes of visual feature $v$, as shown in Figure 6.
> > >
> > > **Practical Scalability:** To directly answer the reviewer's question from their original review (i.e. "Can the author extends the experiments from #404 10k subset to a much larger set, like 500k?"), we had extended our real-world evaluations of Symbal to 100K image-text pairs from ShareGPT4V in our previous response.  By successfully identifying nontrivial systematic errors at this scale - 20x larger than the standard COCO-Val set - we have empirically shown that Symbal does not require any "prior partitions" and can scale reliably.
> > >
> > > From an algorithmic perspective, Symbal relies on embedding-based clustering, image-text scoring, and summarization; each of these tasks is optimized via libraries like Faiss and vLLM. There is no algorithmic bottleneck in Symbal that would prevent it from scaling to the million- or billion-scale given adequate compute.
> > >
> > > We hope that the responses above answer your questions, and we're happy to provide additional clarifications if necessary.

---

### Decision · Program_Chairs · 2026-04-30

**Decision:**

Accept (regular)

**Comment:**

Summary:

This paper introduces Symbal, a framework that uses off-the-shelf VLMs to identify systematic misalignment in image-caption pairs at the dataset level and to summarize the results in natural language. To demonstrate the effectiveness of Symbal, the authors created SymbalBench, a benchmark comprising 420 visual-language dataset settings across two domains: natural and medical images. The experiments show that Symbal correctly identifies misalignment in 63.8% of the dataset settings, a 4x improvement over the closest baseline. The authors also demonstrate that Symbal is effective in detecting misalignment in VLM-generated datasets such as ShareGPT4V.

Justifications:

The proposed framework and benchmark have the potential to improve the quality of image-caption datasets, as demonstrated by the misalignment detection in ShareGPT4V. Three reviewers are in consensus to recommend a weak acceptance for this paper. There are still concerns from reviewers QUk2 and PLXF after the rebuttal. I recommend that the authors incorporate all reviewers' feedback and additional experiments into the final version during the rebuttal. During the rebuttal, 1.7 million image-caption pairs from 420 datasets were mentioned, but this fact was not mentioned in the original submission.